# Decadal and spatially complete global surface chlorophyll-a data record from satellite and BGC-Argo observations

Daniel J. Ford [1], Gemma Kulk [2,3], Shubha Sathyendranath [2,3] and Jamie D. Shutler [1]

[1] Centre for Geography and Environmental Sciences (CGES), University of Exeter, Penryn, UK.
[2] Marine Processes and Observations, Plymouth Marine Laboratory, Prospect Place, The Hoe, Plymouth PL1 3DH, UK
[3] National Centre for Earth Observation, Plymouth Marine Laboratory, Prospect Place, The Hoe, Plymouth PL1 3DH, UK

*Correspondence to*: Daniel J. Ford (d.ford@exeter.ac.uk)

**Abstract.** Decadal-scale satellite-based climate data records of chlorophyll-a (chl-a), an essential climate variable, are now readily available at high accuracy and precision. These data are being extensively used for research and, increasingly, for operational services. However, these observations rely on availability of sunlight and the satellite sensor being able to view the ocean, so there are gaps in data due to the presence of clouds and more widely during the polar winter. This is an issue when spatially complete data are needed for global climate studies, or as inputs to machine learning methods and for data assimilation. Whilst addressing cloud cover is well studied, methodologies to overcome missing data due to the polar winter has received little attention and simple approaches to overcome these gaps can lead to unrealistic values. Biogeochemical Argo (BGC-Argo) floats have widely been deployed, and they represent an opportunity to address these gaps. We present an approach that combines BGC-Argo data and a satellite chl-a climate data record to produce a spatially and temporally complete, global monthly chl-a record between 1997 and 2024 at 0.25º spatial resolution. Clouds gaps were filled using an established spatial kriging approach. Polar wintertime chl-a were reconstructed using relative changes between the wintertime BGC-Argo chl-a, and the previous autumntime or next springtime satellite observations, for individual hemispheres. Uncertainties were calculated on a per-pixel basis to retain the underlying uncertainty fields in the climate data record and were modified to account for the uncertainties related to the gap filling. The seasonal cycles in the resulting polar data are consistent with light availability. Clear interannual and inter-hemisphere variability in the wintertime chl-a were observed. Independent assessment of solely the gap filled wintertime chl-a estimates against *in situ* data (N = 201 total) indicates that the accuracy and precision of the underlying satellite data, a key component of a climate data record, are maintained. The 26 year global and spatially complete chl-a data, that are consistent with the underlying climate data record can be downloaded from Zenodo (Ford et al., 2025b).

## 1. Introduction

Chlorophyll-a (chl-a), the dominant photosynthetic pigment in phytoplankton, has been identified as an essential climate variable by the Global Climate Observing System for assessing current and future changes to oceanic global bio-

geochemical cycles (GCOS, 2021). Satellite-ocean-colour-based synoptic chl-a fields of the surface and near-surface ocean (varying from a few millimetres depth to tens of metres dependent upon the water constituents) have routinely been produced since the launch of Sea-viewing Wide Field-of-view Sensor (SeaWiFS) in September 1997 and the more advanced satellite ocean colour sensors that have followed. The ocean colour signal at different wavelengths of light can be related to *in situ* chl-a concentrations and used to estimate synoptic scale chl-a (e.g. Gohin et al., 2002; Hu et al., 2012; O'Reilly & Werdell, 2019). These observations from multiple satellites that cover different time periods and often with different sensor characteristics, are now routinely merged into continuous climate data records, the main effort of which results in the Ocean Colour Climate Change Initiative (OC-CCI) (Sathyendranath et al., 2019). These records are essential for assessing global and regional changes in phytoplankton abundance and primary production (Kulk et al., 2020). Additionally they are routinely used for ecosystem monitoring, understanding biogeochemistry, supporting fisheries management, water quality monitoring and operational ocean forecasting (Sathyendranath et al., 2023b).

However, ocean-colour observations of chl-a have the limitation of data gaps due to cloud cover, and high sun zenith and viewing angles which routinely occur during polar winter. Multiple methods have been developed to fill the data gaps due to cloud cover on both regional and global scales. For example, Saulqiun et al. (2018) used an optimum interpolation technique (commonly used in sea surface temperature records) to gap fill a merged operational chl-a record. Liu and Wang (2018) used the Data Interpolating Empirical Orthogonal Functions (DINEOF), which reconstructs the missing chl-a based on empirical orthogonal functions, to fill Visible Infrared Imaging Radiometer Suite (VIIRS; one of the inputs to the multi-sensor OC-CCI record) observations. Recently, Hong et al. (2023) used a convolution neural network that ingests environmental information, including sea surface temperature and photosynthetically active radiation, to aid in the reconstruction of chl-a underneath clouds between ~50ºN and ~50ºS within the OC-CCI record. These methods show differing accuracies, but they are generally effective at reconstructing gaps due to cloud cover (Stock et al., 2020). However, none of these studies attempt to reconstruct the persistent gaps at high latitudes that occur during the polar winter.

These missing high-latitude polar winter data often make the exploitation of the overall chl-a record more challenging or result in assumptions being made about the missing wintertime chl-a concentrations. For example, within efforts to reconstruct the global ocean carbon dioxide ($CO_2$) sink, these missing data are often manually filled with a fixed value (e.g., Chau et al., 2022; Gregor & Gruber, 2021), or the gaps mean that chl-a data are avoided for the input variables used to interpolate other data which means that any explicit biological signal within the interpolation is omitted (Ford et al., 2024a). Example fixed values include Gregor and Gruber (2021) who use a fixed value of 0.3 mg m$^{-3}$ for all missing polar wintertime data, whereas Chau et al. (2022) use a value of 0.0 mg m$^{-3}$. These practical choices likely influence the underlying interpolation and reconstructions of the data (in this case the ocean $CO_2$ sink) and are unlikely to be scientifically applicable across all times and geographic locations as they overlook regional and temporal variations and create unnatural boundaries or characteristics (e.g., the Arctic and Southern Ocean likely have different bio-geochemical characteristics). The

expanding availability of autonomous BGC-Argo profilers with chl-a sensors (Roemmich et al., 2019) that collect observations within the polar winter provides an opportunity to generate a data-driven reconstruction of these missing wintertime chl-a. Whilst able to provide data during polar winter, the BGC-Argo chl-a data have reduced accuracy with respect to the satellite observations, so using them to directly gap-fill the higher accuracy climate data record presents some challenges.

We present a methodology for producing a spatially complete monthly chl-a record between October 1997 and December 2024 at a spatial resolution of 0.25° with spatially resolved uncertainties. Cloud gaps were initially filled using an established spatial kriging approach. The missing polar wintertime chl-a data were then filled using relationships between BGC-Argo measured chl-a and the first spring or last autumntime observations within the satellite record. Relative changes between the satellite record and BGC-Argo chl-a were used rather than relying on the absolute values to overcome the difference in depth relevance of the BGC-Argo chl-a versus the satellite record. The relationships were constructed for both the Northern Hemisphere (which includes all of the Arctic Ocean) and Southern Hemisphere (containing the Southern Ocean) separately, which highlights the bio-geochemical differences evident in these different wintertime chl-a response.

## 2. Data and Methods

### 2.1. BGC-Argo Chl-a data

BGC-Argo profilers have been deployed globally, and at the time of writing ~400 (~60 %) of these have fluorometric chl-a sensors. Delayed mode BGC-Argo profile data (2008 to 2024, last ingestion: 8th September 2025) (Argo, 2025) were retrieved from the Argo Global Data Assembly Centers (GDAC) for the Southern Hemisphere (south of 40°S) and the Northern Hemisphere (north of 40°N; Figure 1). These delayed mode profiles have undergone automatic processing, and quality control within the GDAC following Schmechtig et al. (2015) and Schmechtig et al. (2023). For each BGC-Argo profile the quality flagging was applied to only retain the highest quality data (quality flag 2). The mean chl-a concentration was extracted from the first optical depth. The first optical depth was estimated from the diffuse attenuation coefficient at 490 nm ($K_d(490)$) which was determined using the shallowest chl-a observation (shallower than 10 m) and the relationship described in Morel et al. (2007). The mean was calculated in $\log_{10}$ space due to the logarithmic distribution of chl-a (Campbell et al., 2002). Profiles with a mean chl-a less than 0.014 mg m$^{-3}$ were discarded as this value was twice the factory-specified sensitivity of the fluorescence sensors (Long et al., 2024). The spatial and temporal distributions of the resulting chl-a profiles in the Southern and Northern Hemispheres are displayed in Figure 2. The BGC-Argo chl-a data were then gridded (mean in $\log_{10}$ space) into monthly 0.25° composites to match the resolution of the satellite observations described in the next section, using existing publicly available software (Ford et al., 2024b).

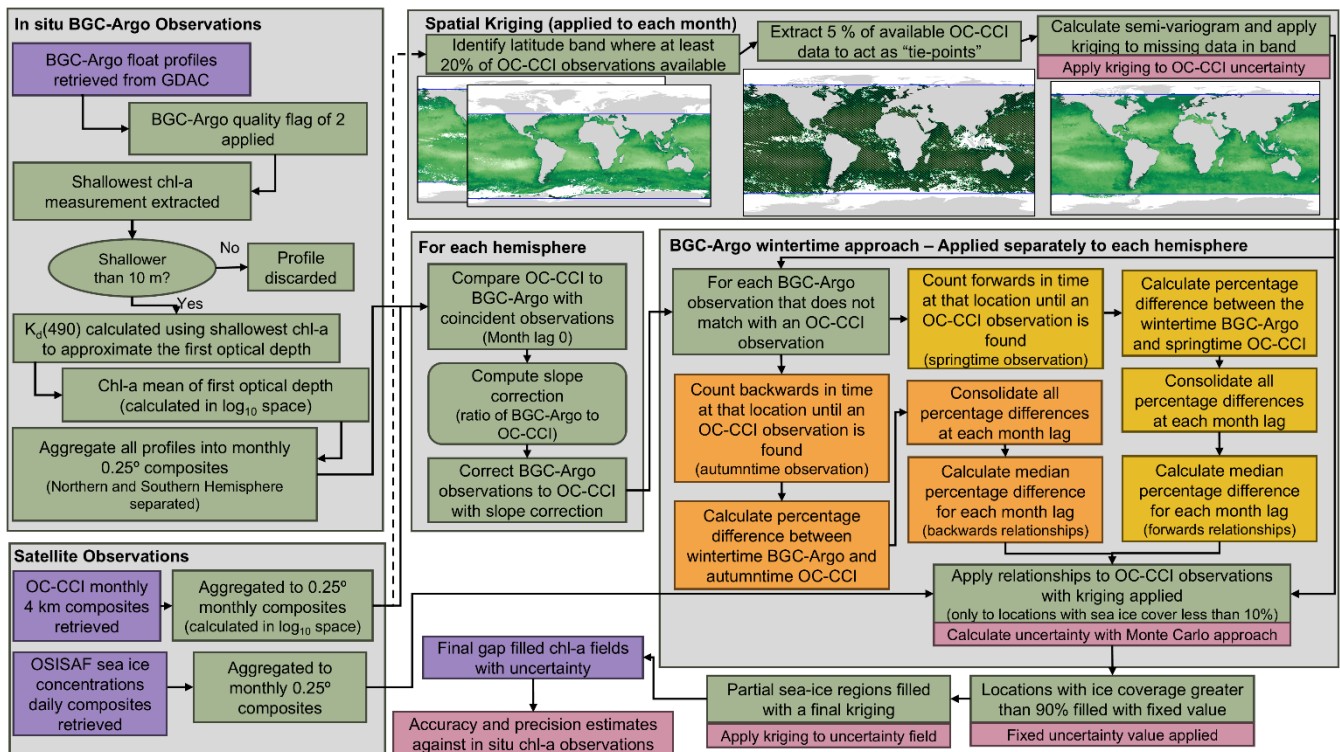

**Figure 1: Schematic showing the methodology for producing the gap filled chlorophyll-a (chl-a) Ocean Colour Climate Change initiative (OC-CCI) record. In flowchart acronyms are Biogeochemical Argo (BGC-Argo), Argo Global Data Assembly Centers (GDAC), diffuse attenuation coefficient at 490 nm ($K_d(490)$) and Ocean and Sea Ice Satellite Application Facility (OSISAF).**

## 2.2. Satellite observational data

The OC-CCI (v6) chl-a data were retrieved at daily and monthly temporal resolution at their native spatial resolution (4 km) between October 1997 and December 2023 (Sathyendranath et al., 2019, 2023a). The process for cross-calibrating and merging the data from multiple satellite ocean colour sensors used within the OC-CCI are described in Sathyendranath et al. (2019). The OC-CCI monthly composites were aggregated after $\log_{10}$ transformation, to compute mean values at 0.25º degree spatial resolution using existing publicly available software (Ford et al., 2024b). The uncertainties (1 sigma; given as the root mean square difference; RMSD) provided with the OC-CCI product are calculated with respect to *in situ* observations within each optical water class (Jackson et al., 2017). These uncertainties were converted to a 0.25º resolution by calculating the mean of the 4 km uncertainties that contribute to each 0.25 grid cell, which assumes spatial uncertainties within adjacent cells are dependent and spatially correlated (Taylor, 1997). Daily sea ice concentration at ~25 km spatial resolution were obtained from Ocean and Sea Ice Satellite Application Facility (OSISAF) (OSISAF, 2022). The daily OSISAF sea ice concentrations were combined into monthly composites and regridded to the same spatial grids as the monthly OC-CCI data, by using the same software (Ford et al., 2024b).

To assess the impact of gridding the BGC-Argo and OC-CCI data, a comparison between the daily OC-CCI at 4km and the individual BGC-Argo chl-a profiles was conducted, following standard ocean colour comparison protocols (Bailey and Werdell, 2006; Ford et al., 2021). Each BGC-Argo profile was matched daily to the OC-CCI record (i.e., coincident day), and the mean chl-a (in $\log_{10}$ space) extracted from a 3-by-3 pixel grid (which represents ~12 km by 12 km region at the equator). The same analysis was then repeated for the monthly 0.25º BGC-Argo and OC-CCI data. A standard suite of statistics was calculated in $\log_{10}$ space for both the daily 4km and the monthly 0.25º data and the results were then compared. The metrics included the bias (accuracy), root mean square deviation (RMSD; precision) and the slope and intercept of a Type II regression. This analysis assessed the impact of averaging the BGC-Argo and OC-CCI observations to monthly 0.25º composites (Supplementary Figure S1) and indicated that the averaging had limited effect on the retrieved unweighted bias (accuracy) and RMSD (precision) for the Southern and Northern Hemisphere (i.e., the bias and RMSD results for the daily matches were similar to the monthly 0.25º gridded data). The high intercept values at both 4 km and 0.25º (Supplementary Figure S1), particularly for the northern hemisphere, illustrate why the absolute values of the BGC-Argo data cannot be used to directly fill the satellite record.

The two sets of observations (i.e., satellite sensor versus Argo), dependent upon the water constituents, overlap in terms of their depth relevance. However, the BGC-Argo chl-a measured by in vivo fluorescence are considered less accurate than the measurement of chl-a by high performance liquid chromatography (HPLC) that is predominantly used for the calibration and evaluation of ocean colour data (Long et al., 2024; Roesler et al., 2017). These differences could lead to an underlying bias between the OC-CCI and the BGC-Argo chl-a observations (e.g., see supplementary Figure S1). To minimise these differences, the BGC-Argo chl-a were corrected with respect to the OC-CCI data where coincident observations were available by using a slope factor correction as outlined in Roesler et al. (2017) (i.e., the median of all individual slope factors for each hemisphere). The Northern and Southern Hemisphere slope factor corrections were 0.916 and 1.967, respectively. These corrections appear initially smaller than those reported in Roesler et al. (2017) and Long et al. (2024), however the delayed mode processing of the BGC-Argo chl-a already includes a slope factor correction of 2. Therefore our slope factors are consistent to the previous work. The slope factor correction does not assume that the OC-CCI record is the 'truth', but our objective is to fill gaps in the OC-CCI data using relative changes to the BGC-Argo chl-a, which requires the two datasets to be consistent.

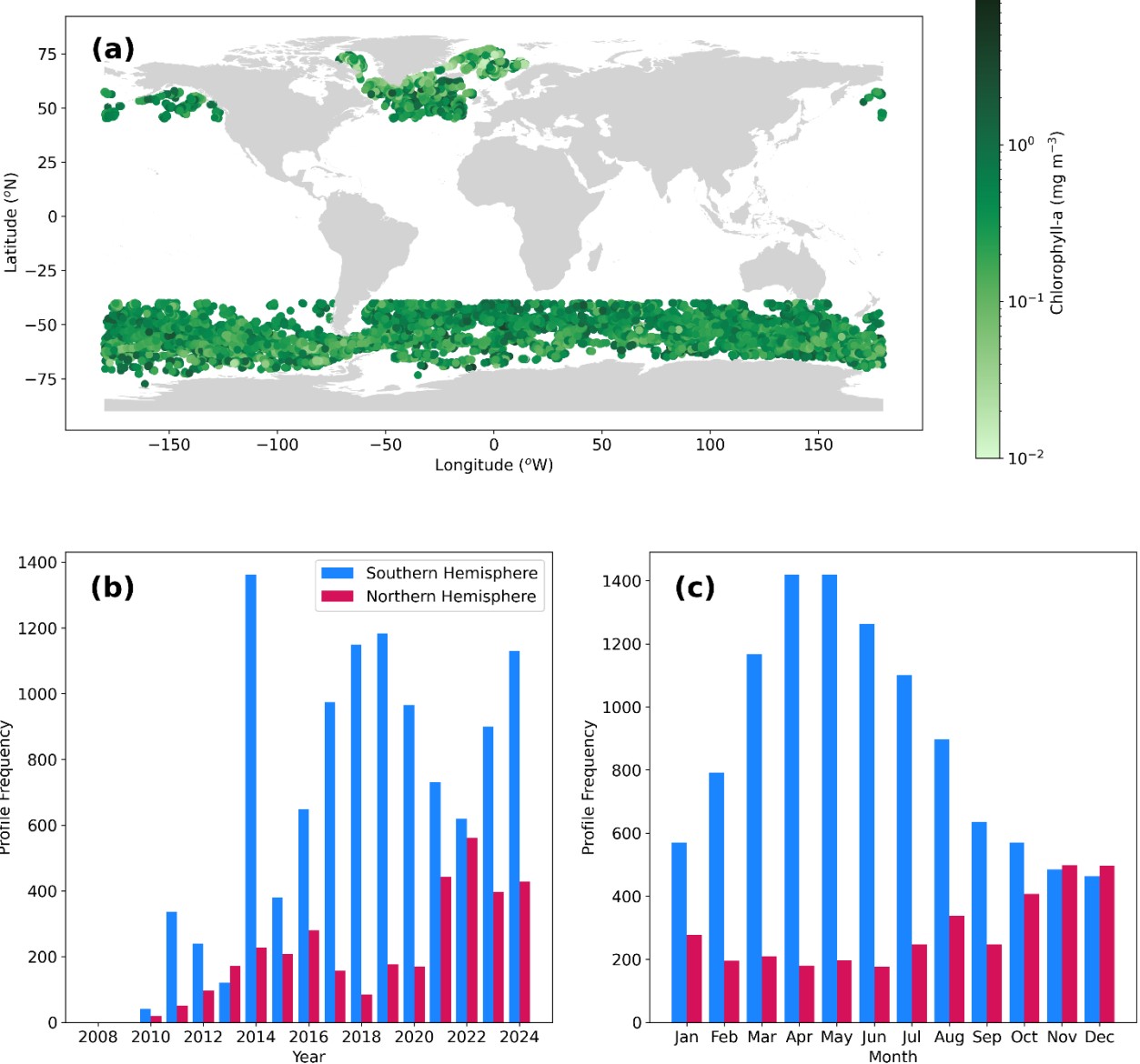

**Figure 2: Spatial and temporal distribution of the BGC-Argo chlorophyll-a profiles used. (a) Geographical distribution of the individual profiles (i.e., individual profilers appear multiple times). (b) Temporal distribution in years of the number of profiles used for the Southern Hemisphere (<40ºS) and Northern Hemisphere (>40ºN). (c) Temporal distribution in months of the profiles used for the Southern Hemisphere (<40ºS) and Northern Hemisphere (>40ºN). Basemap in (a) from Natural Earth v4.0.0 (https://www.naturalearthdata.com/).**

## 2.3. Spatial Kriging for cloud gap filling

Ocean-colour sensors on board polar orbiting satellites collect data at multiple wavebands in the visible domain, which is used to estimate the chl-a concentration (e.g., Gohin et al., 2002; O'Reilly and Werdell, 2019). Clouds are optically thick in the visible spectrum, and so they block the sensor's view of the ocean, leading to missing data within the ocean-colour chl-a data record. The use of monthly composites of ocean-colour-based chl-a reduces the gaps due to cloud cover as clouds tend to evolve (i.e., appear to move) faster than the ocean conditions. Multiple daily observations within the one-month allows the impact of the faster moving clouds to be averaged out. But aggregating data over multiple days cannot help in regions where clouds can be more prevalent, such as the inter-tropical convergence zone, or in regions where other features, such as coccolithophore blooms, inhibit the satellite retrieval of chl-a. Stock et al. (2020) evaluated multiple approaches to fill data gaps due to cloud cover, suggesting that approaches including spatial kriging or DINEOF were the most accurate. More complex approaches could be implemented such as Optimum Interpolation or spatio-temporal kriging, but these, as well as DINEOF, come with increased computational requirements. Here we implement a spatial ordinary kriging approach to fill cloud cover (or other) gaps (Figure 1) as the use of monthly composites in this study reduces the number of data gaps, and therefore the computational requirements of more complex approaches are unlikely to improve the estimates further.

To perform the kriging, a semi-variogram was computed for each monthly timestep in the timeseries using SciKit-Gstat v1.0.0 (Mälicke, 2022) with the 'martheon' estimator and an exponential function. A visual inspection of the semi-variogram output was used to optimise the estimator and function. The semi-variogram was fit to a ~5% subset of the OC-CCI observations that were equally distanced in space, for a monthly varying latitude band (e.g., 50 ºN to 80 ºS; Figure 1) where at least 20% of the OC-CCI observations are available. The subset size was a computational choice because the number of pairwise distances that must be calculated by the semi-variogram is a $n^2$ function, where $n$ is the number of locations. Setting the monthly varying latitude limits (i.e latitude band where at least 20 % of the OC-CCI observations are available) prevents the kriging from filling data that are missing due to the polar winter and not due to cloud cover or other features (Figure 1). The ordinary kriging was applied only to the missing data locations (i.e., the original OC-CCI observations are left unchanged) and was set to use the nearest six observations to fill a missing data location to limit the influential distance of each observation.

## 2.4. BGC-Argo Wintertime filling

The approach developed with the BGC-Argo profilers to reconstruct the wintertime observations is based on the assumption that wintertime chl-a will decline due to lower light availability before then increasing again as the light returns in the spring. Therefore, the wintertime chl-a would be lower than the last available OC-CCI observations in autumntime and first available observations in springtime. Thus, the decline in chl-a during the polar winter can be estimated using the BGC-Argo profiler chl-a as an observational constraint.

For each BGC-Argo observation, we count backwards in time to the last autumntime observation up to nine months prior to the BGC-Argo observation time (Figure 1). For example, a time lag of zero indicates a coincident OC-CCI observation with the BGC-Argo, and a lag of one month indicates the OC-CCI observation occurred in the previous month to the BGC-Argo. The percentage difference was calculated between the OC-CCI and bias-corrected BGC-Argo chl-a in mg m$^{-3}$ (to avoid the switch from positive to negative values within the $\log_{10}$ transformed values). All of the available percentage differences at

each time lag were consolidated and the median of all the individual percentage differences were calculated. This constructs a median relative change relationship between the autumntime OC-CCI and BGC-Argo chl-a observations in terms of the time to the OC-CCI observation (i.e., the median of all the percentage differences between 2010 to 2024 at each time lag; Figure 2). The procedure was applied to the Southern and Northern Hemispheres separately, which allows for the known biogeochemical differences between the two polar regions (Ardyna and Arrigo, 2020; Deppeler and Davidson, 2017). As the

cloud gap filling approach was applied before this step, the "OC-CCI observation" is likely to be an original OC-CCI retrieval but may in some instances be a cloud-filled value. Although the process was applied up to nine months back in time, in practice the majority of data are filled within ± 4 month (Supplementary Figure S2).

The whole procedure was then applied in reverse, counting forwards to the first springtime observation. This provides both a

backwards looking and forwards looking relationship over the wintertime period for each hemisphere. The obtained relationships between OC-CCI and BGC-Argo data were then used to gap fill the gridded wintertime OC-CCI data. For each wintertime pixel, the time lag between the autumntime and springtime OC-CCI observations was calculated, and the relationship with the lowest time lag was used (i.e., either the forward or backward in time relationship). If both had the same time lag, the autumntime relationship was used in preference as these were generally constrained by more BGC-Argo

observations (N = 1072; Figure 3a) than the springtime relationship (N=719; Figure 3b).

Chl-a concentrations in regions with a sea ice coverage greater than 95 % as indicated by the OSISAF sea ice concentrations were set to a fixed value of 0.1 mg m$^{-3}$(Figure 1). We selected this value from literature (e.g. Boles et al., 2020; Randelhoff et al., 2020), based on wintertime chl-a concentrations from under ice regions that are not experiencing an significant or

enhanced under ice phytoplankton growth, but we acknowledge the potential for highly heterogenous under ice chl-a concentrations.

Any remaining pixels that were not gap filled by any of the previous procedures are filled with a final kriging pass, following the same methodology as in section 2.3, but globally, which was applied to ~3% of the data (Supplementary Figure S3). This

final step was mainly applied in regions of partial sea ice coverage (i.e., those with ice coverage between ~10 and 95%; Figure 1). A breakdown of the area filled by each stage in the gap filling methods are given in Supplementary Figure S3, and the BGC-Argo relationship month lags are given in Supplementary Figure S2.

An implication of applying the BGC-Argo gap filling approach to monthly resolution data leads to artificial latitude banding due to the month lag relationship changing at each latitude. This banding in the monthly gap-filled record is highlighted here as it is dependent upon the methodological choices and data limitation issues. The month lag relationships could instead be linearly interpolated and applied to 8-day composites and then averaged to monthly composites which would likely reduce this banding. But constructing the BGC-Argo relationships using 8-day composites is currently not feasible due to limited BGC-Argo data availability, especially at the higher time lags.

## 2.5. Uncertainty propagation (1 standard deviation; 1σ)

The cloud gap filling kriging approach uses observations in the vicinity of the empty pixel (based on the semi-variogram described in section 2.3) to construct the missing chl-a. The uncertainties arising from using the ordinary kriging are a combination of: (1) the underlying OC-CCI uncertainties in the measurements and (2) those arising from the method used to estimate the missing chl-a. Therefore, the accompanying OC-CCI uncertainty fields were also kriged using the same semi-variogram estimated from the chl-a observations producing uncertainty values for each gap filled pixel.

The polar wintertime data filled with the BGC-Argo relationships use the spring- and autumntime observations which are then multiplied by the percentage reduction in chl-a. Therefore, two sources of uncertainty combine to form the total uncertainty: (1) the uncertainty in the OC-CCI spring- and autumntime observations (1σ), and (2) the uncertainty in the percentage difference estimated from the BGC-Argo profiler (1σ). We estimate the uncertainty in the percentage difference by calculating the median absolute deviation (MAD) and convert this to a standard deviation equivalent with the scaling factor of 1.4826 (Rousseeuw and Croux, 1993). Using the MAD reduces the sensitivity to "outliers" within the percentage differences. Both sources of uncertainty were propagated through the analysis using a Monte Carlo uncertainty approach with 1,000 ensembles, assuming they are independent and uncorrelated. Each source of uncertainty was propagated by randomly perturbing the input value (i.e., the percentage difference and OC-CCI chl-a observation) using a random number generator that produces a normal distribution with a standard deviation defined by the uncertainty. The wintertime chl-a was then recalculated for each perturbed input in the ensemble. The standard deviation of the 1,000 ensembles was taken as the uncertainty and the resulting spatially varying uncertainty were provided in $\log_{10}$(mg m$^{-3}$) units for each of the polar wintertime filled pixels (to be consistent to the underlying OC-CCI record).

The under ice chl-a uncertainty was set to 0.4 $\log_{10}$(mg m$^{-3}$), owing to the complex dynamics under sea ice based on a range of sources (Ardyna and Arrigo, 2020; Arrigo et al., 2012, 2014; Boles et al., 2020; Randelhoff et al., 2020) – see the discussion section for further details on this decision. The uncertainties are provided alongside the gap-filled chl-a data, providing consistent spatially and temporally varying uncertainties.

## 2.6. Independent accuracy and precision evaluation

To independently assess the accuracy and precision of the gap filled wintertime chl-a concentrations we used the OC-CCI chl-a validation dataset (Valente et al., 2022; v3). These data provide the chl-a concentration measured on ships by either HPLC or fluorometric approaches. This dataset is used routinely to assess the accuracy and precision of the OC-CCI record (but are not used to tune the algorithms used). Furthermore, the wintertime values remain independent as these cannot be matched to the original OC-CCI record as there are gaps in the original satellite data record. The individual chl-a observations were quality controlled with the flags provided within the dataset and the chl-a was retained if greater than 0.01 mg m$^{-3}$. The retained chl-a observations were gridded onto the same monthly 0.25 ° grid (mean in $\log_{10}$ space) as the gap-filled OC-CCI record, separately for the HPLC and fluorometric chl-a observations. The gridded *in situ* observations were then compared with the gap-filled OC-CCI data record at the locations where the BGC-Argo relationships were applied (i.e the polar wintertime filled data) for the Northern Hemisphere and Southern Hemisphere, separately. The standard suite of statistics described in section 2.2 were calculated to assess the accuracy and precision of the gap filled wintertime chl-a against these independent *in situ* observations.

## 3. Results

The relationships between the autumntime (backwards; Figure 3a, c) and springtime (forwards; Figure 2b, d) OC-CCI and wintertime BGC-Argo profilers showed clear differences between the Southern (Figure 3a, b) and Northern Hemisphere (Figure 3c, d). The Southern Hemisphere indicated a slower decline in the chl-a concentration from the previous autumntime compared to the Northern Hemisphere. For example, the one-month lag showed a median 24% decrease in chl-a for the Southern Hemisphere compared with a median 69% decrease for the Northern Hemisphere. The springtime comparison also showed similar regional differences (Figure 3b). The forward relationships both indicate similar percentage differences for month lags 1 to 5. The backward and forward relationships at month lags greater than 5 showed more variability, which is likely due to the lower number of available data to construct the relationships (Figure 3).

Applying both the spatial kriging to fill the cloud gaps, and then BGC-Argo approach to fill the wintertime polar chl-a, allowed for the production of a globally complete observation-based gap-filled chl-a dataset (Figure 4). Using monthly composites reduced the need to broadly apply the cloud gap filling approach and it was mainly applied in earlier years of the timeseries when SeaWiFS was the only ocean colour satellite available (1997-2002; see Supplementary Figure S3). Focusing on further analysis of polar regions where the BGC-Argo wintertime approach was applied, four exemplar locations showed regional differences in the chl-a wintertime concentrations (Figure 4a,b, d, e). The selected locations in the Northern

Hemisphere (Figure 4a, b) generally showed larger decreases in the wintertime chl-a, although concentrations in the North Atlantic Ocean (Figure 4a) had a much larger decline than in the North Pacific Ocean (Figure 4b). The Southern Hemisphere also showed regional difference across the three selected locations (Figure 4d, e). These timeseries highlight that the approach was able to produce a consistent timeseries that captured interannual variability in the wintertime chl-a (e.g., particularly evident in Figure 4d, e). Condensing the full timeseries into a multi-year monthly climatology illustrates the full coverage and spatial and temporal variability of the wintertime period that has been filled by the BGC-Argo approach (Figure 5). These multi-year monthly climatologies show the advantage between using only OC-CCI observations and the gap-filled data, especially at the beginning and end of the winter period, where the cloud gap filling approach aids in reconstructing the seasonality (Figure 5b, d, e). They also reinforce the regional differences in the magnitude of the chl-a decline during winter.

Total uncertainties in the wintertime chl-a were ~0.6 $\log_{10}$(mg m$^{-3}$) in the Northern Hemisphere (Figure 5) where this value is driven mainly by the uncertainty in the relationship determined via the BGC-Argo profilers. The higher uncertainty in the relationship was a combination of fewer Argo profiles, which in turn effects the constraint of the complex wintertime chl-a response in the Northern Hemisphere (Figure 4a, b). The Southern Hemisphere has lower total uncertainties of ~0.4 $\log_{10}$(mg m$^{-3}$) (Figure 5) that contained more equal contributions to the uncertainties from the BGC-Argo relationship and the underlying spring- and autumntime OC-CCI chl-a uncertainty. The accuracy and precision of the wintertime chl-a (ie the polar gap filled data) was assessed against independent *in situ* chl-a observations, determined either by HPLC (Figure 6a, c) or fluorometric methods (Figure 6b, d). Comparisons to the *in situ* HPLC chl-a concentrations show higher numbers of coincident observations for the Northern Hemisphere (N = 136; Figure 6a) compared with the Southern Hemisphere (N = 65; Figure 6c). Both hemispheres show consistent accuracy and precision between the *in situ* observations and wintertime chl-a, with small biases (less than 0.1 $\log_{10}$(mg m$^{-3}$)) and RMSD of ~0.4 $\log_{10}$(mg m$^{-3}$). Comparison with the *in situ* fluorometric chl-a resulted in insufficient coincident observations to draw conclusions for the Southern Hemisphere (N = 11; Figure 6d). For the Northern Hemisphere, the accuracy of the wintertime chl-a is consistent to the previous comparisons to the BGC-Argo chl-a displaying the same regional biases (N = 875; Figure 6b; Supplementary Figure S1 d). Although, the precision is larger as highlighted by the RMSD values ~0.6 $\log_{10}$(mg m$^{-3}$) compared to ~0.4 $\log_{10}$(mg m$^{-3}$) for the comparison to the BGC-Argo chl-a (Supplementary Figure S1 d). These precision estimates for the Southern Hemisphere (~0.4 $\log_{10}$(mg m$^{-3}$); Figure 6c) and Northern Hemisphere (~0.6 $\log_{10}$(mg m$^{-3}$); Figure 6b) support the validity to the propagated uncertainty estimates for the Southern Hemisphere of ~0.4 $\log_{10}$(mg m$^{-3}$) and the Northern Hemisphere of ~0.6 $\log_{10}$(mg m$^{-3}$). In the Southern Hemisphere these uncertainties are of similar magnitude with the OC-CCI uncertainties for the region, however they are larger within the Northern Hemisphere.

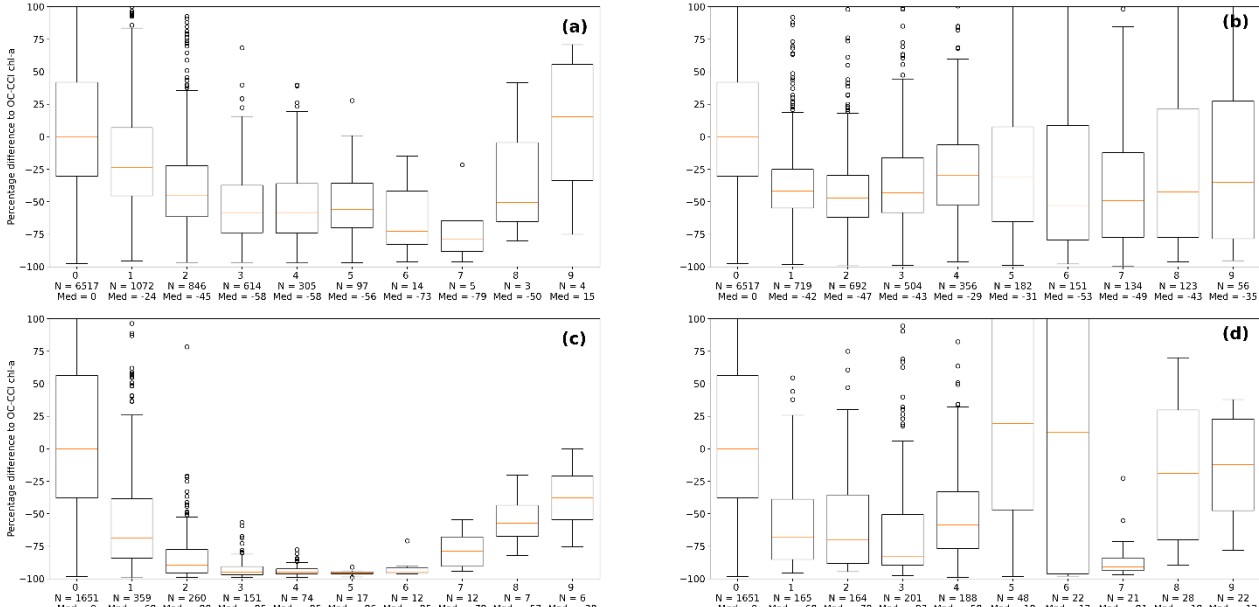

Figure 3: (a) Boxplots indicating the percentage difference between the Ocean Colour Climate Change Initiative (OC-CCI) and slope-corrected BGC-Argo chlorophyll-a (chl-a) based on time lag since the last autumntime observation in the Southern Hemisphere. Red line indicates the median, the box indicates the 25% and 75% quartiles, and circles indicate data considered as outliers. In axis abbreviations are number of samples (N) and median bias (Med). (b) same as (a), but for the Southern Hemisphere springtime relationship. (c) same as (a), but for the Northern Hemisphere autumntime relationship. (d) same as (a), but for the Northern Hemisphere springtime relationship. Y-axis limits have been selected to emphasise the lower time lags, where the higher time lags are constrained by less data and have limited use in the methodology.

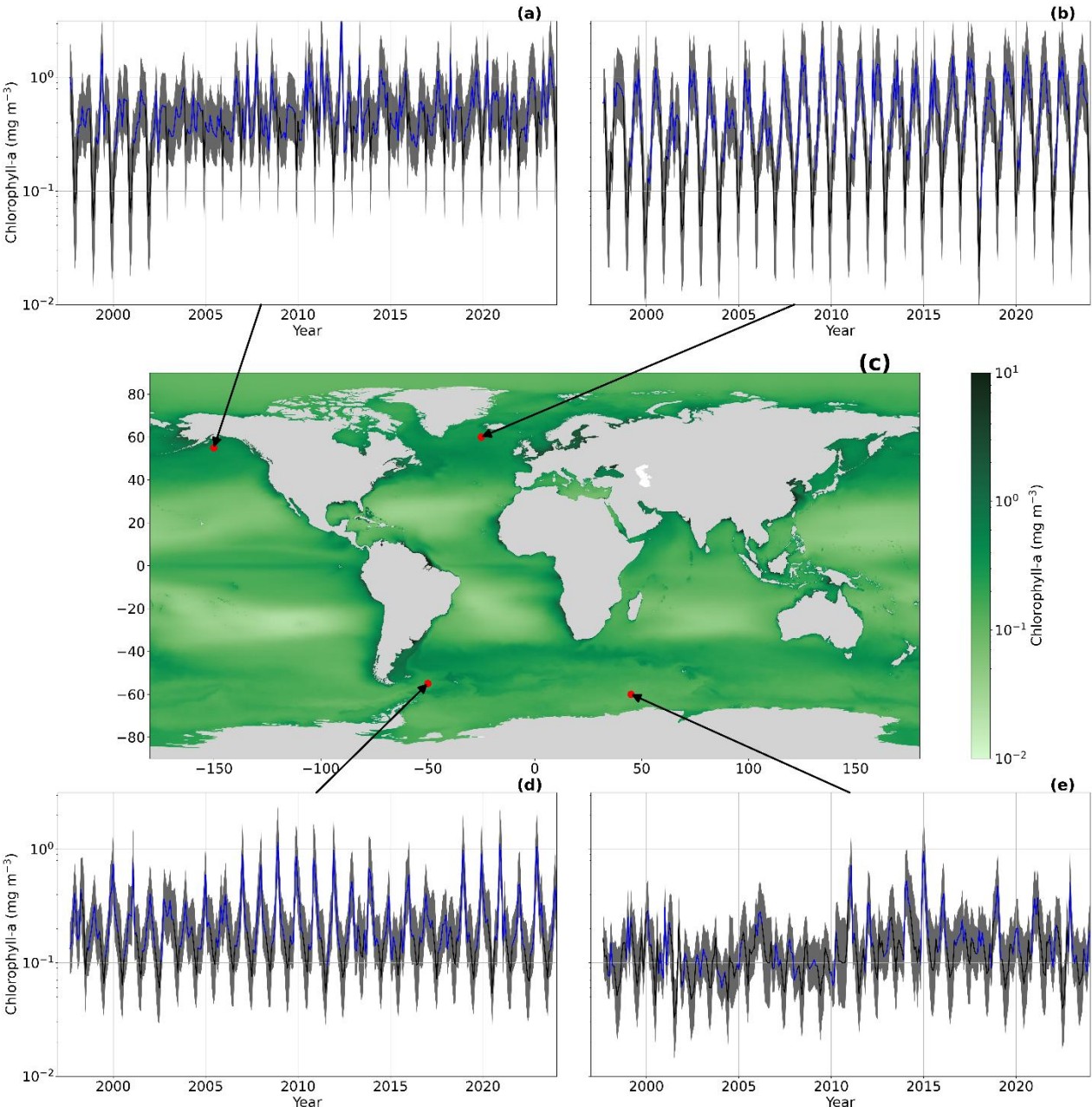

**Figure 4: (a) Chl-a timeseries extracted for the location marked by the arrow between 1997 and 2024 (plotted using consistent y axes). Blue line indicates the OC-CCI timeseries without gap filling and black lines indicate the gap filled data. Grey shaded region indicates the 1σ uncertainty in chlorophyll-a. (b), (d), (e) same as (a) but for their respective locations. (c) Ocean Colour Climate Change Initiative (OC-CCI) chlorophyll-a (chl-a) with gap filling approach applied to the full timeseries climatology (1997 – 2024). We note that areas with a sea ice coverage greater than 90% are set to a fixed value of 0.1 mg m⁻³. Basemap in (c) from Natural Earth v4.0.0 (https://www.naturalearthdata.com/).**

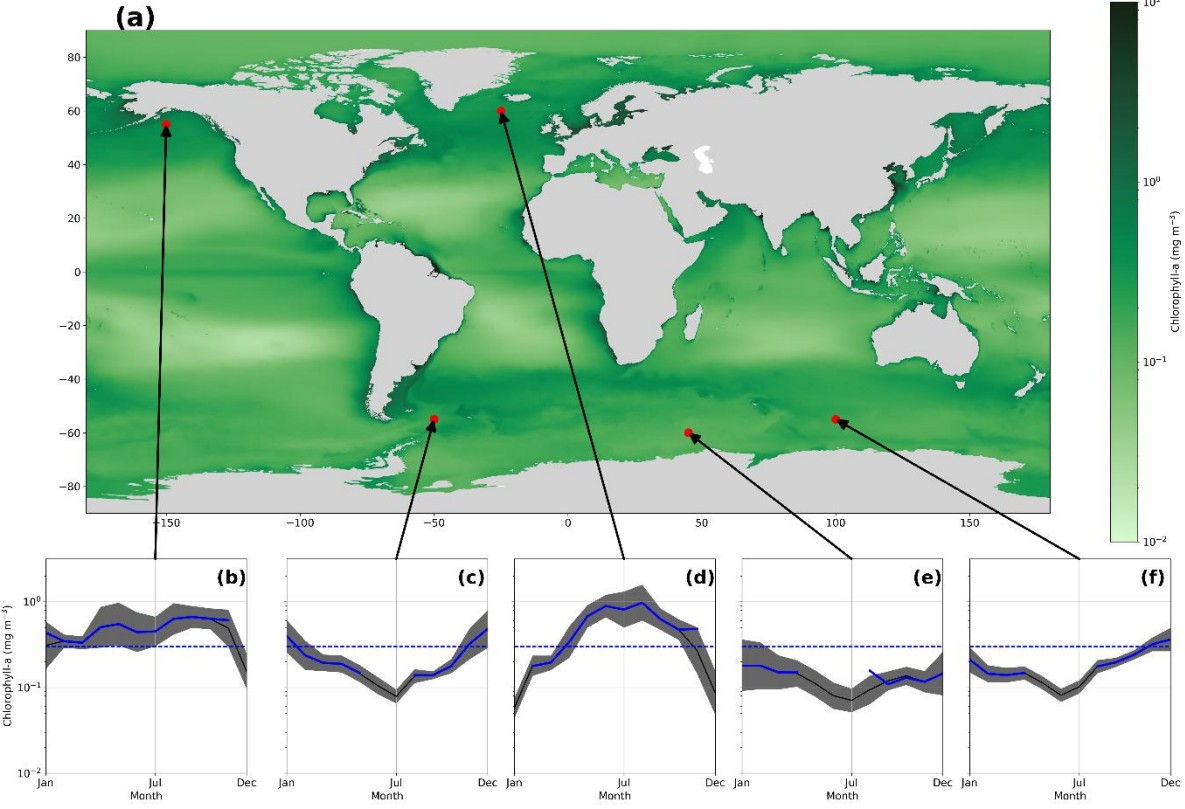

**Figure 5: (a) Ocean Colour Climate Change Initiative (OC-CCI) chlorophyll-a (chl-a) with gap filling approach applied to the full timeseries climatology (1997 – 2024). We note that areas with a sea ice coverage greater than 90% are set to a fixed value of 0.1 mg m$^{-3}$. (b) Monthly climatology calculated at the location marked by the arrow. Blue line indicates the monthly climatology for the OC-CCI timeseries. Black line indicates the monthly climatology for the gap-filled OC-CCI, where the grey shading indicates one standard deviation of the gap-filled climatology. Dashed blue line indicates a chl-a value of 0.3 mg m$^{-3}$ and is referred to in the text. (c), (d), (e) and (f) same as (b), but for their respective locations. Basemap in (a) from Natural Earth v4.0.0 (https://www.naturalearthdata.com/).**

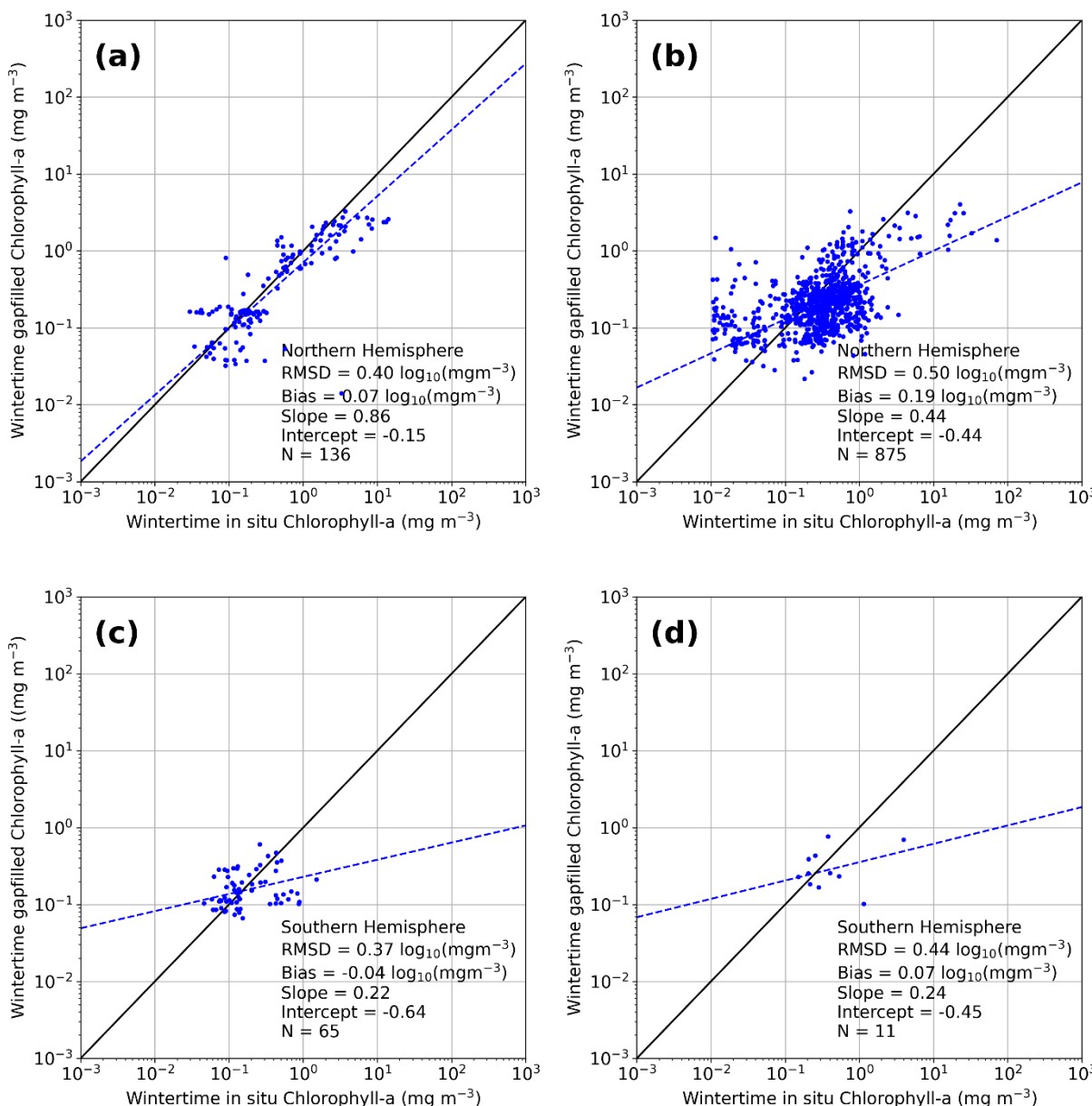

**Figure 6: (a) Comparison between monthly 0.25°-gridded *in situ* HPLC chl-a concentrations and the wintertime gap-filled chl-a data in the Northern Hemisphere. Solid line is 1:1 and dashed blue line indicates a Type-II linear regression. In text the abbreviations for the statistical measures are root mean square difference (RMSD) and number of samples (N). (b) Comparison between monthly 0.25°-gridded *in situ* fluorometric chl-a concentrations and the wintertime gap-filled chl-a in the Northern Hemisphere. (c) Same as (a), but for the Southern Hemisphere. (d) same as (b), but for the Southern Hemisphere.**

340

## 4. Discussion

The exploitation of satellite-based chl-a data records can be hampered by missing data due to cloud cover and missing data during the polar winter due to low solar elevations. In this study, we have presented an observation-based approach to gap filling the missing polar wintertime chl-a data within a satellite climate data record, using the relative change in BGC-Argo profiler chl-a data as an observational constraint. This process has been preceded by using a relatively simple kriging approach to fill missing data due to clouds or other features (such as coccolithophore blooms) that limit the chl-a retrieval. This kriging is then used again as a final step (after the polar data have been filled) to fill any remaining missing data (generally in marginal sea ice zones) which ensures an ocean-colour-based chl-a data record with full global coverage. The cloud gap filling based on a spatial kriging approach used in this study could be regarded as a simple method. Stock et al. (2020) showed that spatial kriging performed well for cloud gap filling, when compared to more complex methods, such as DINEOF. In this study the use of monthly composites does reduce the number of observations that need to be gap-filled by the cloud kriging approach (Supplementary Figure S3), and therefore the computational cost of more complex methodologies likely outweighs any benefit to the retrieved chl-a. This situation is confirmed as the chl-a in regions where gaps were filled using the spatial kriging approach showed good performance with respect to the independent in situ observations (Supplementary Figure S4). Applying the full methodology to generate a higher-temporal resolution dataset (e.g., by using 8-day composites instead of monthly composite) is possible but could present challenges when larger regions are covered by clouds. This may suggest that more complex methodologies, such as those proposed by Hong et al. (2023), using a convolution neural network (that considers the physical and biological conditions), could be more applicable to filling larger cloud cover gaps. With these larger gaps, the computational cost of these more complex methodologies could be beneficial in improving the retrieved chl-a.

It is important to note that spatial gap filling methods make assumptions about the missing data and use chl-a observations from clear sky conditions to fill these gaps. This will likely lead to an underestimation of chl-a concentrations due to photoacclimation by phytoplankton under reduced light from persistent cloud cover (i.e., increasing intracellular chl-a due to lower light conditions) (Begouen Demeaux et al., 2025). The construction of monthly composites of chl-a from observations in clear sky conditions could lead to a varying underestimation of chl-a based on regional cloudiness, for example subtropical gyres are likely less effected due to persistent atmospheric high pressure. This limitation is not unique to this study as it will affect any ocean colour chl-a data product (e.g., Hong et al., 2023; Saulquin et al., 2018) and we therefore consider this outside the scope of the present study.

The data gaps in optical satellite data due to the polar winter have so far received little attention within gap filling methodologies. The approach within this study provides an observation-based gap filling that exploits the expected underlying temporal signal due to light availability and the fundamental requirement for biological growth to need light. The

results appear consistent with previous studies of wintertime chl-a variability. For example, Randelhoff et al. (2020) showed a decline in chl-a to ~0.03 mg m$^{-3}$ in January in Baffin Bay in the Arctic Ocean. We showed wintertime chl-a consistent to these observations (Figure 3d, 4d) during January. Ko et al. (2024) showed wintertime values of ~0.15 mg m$^{-3}$ in the Chuckhi Sea, which would be consistent with the observations near the shelf in the North Pacific Ocean (Figure 4a, 5a). These comparisons to earlier results in the literature indicate that the BGC-Argo relationships applied to the OC-CCI data are able to capture the wintertime chl-a concentrations and their regional differences. The independent assessment conducted using the OC-CCI in situ chl-a validation dataset (Valente et al., 2022) in this study reinforces that the wintertime chl-a values are consistent with the observations (Figure 6) and that they maintain any potential regional biases in the underlying OC-CCI dataset. However, both the comparisons to literature and the validation dataset are limited by the number of *in situ* measurements collected during winter.

The decline in chl-a during winter as identified by the BGC-Argo relationships indicate clear differences between the Southern and Northern Hemisphere (Figure 3). This appears to be consistent with our understanding of biogeochemical differences between the two regions (Arteaga et al., 2020; Deppeler and Davidson, 2017). For example, the Southern Hemisphere showed a slower decline in chl-a during the winter compared with the Northern Hemisphere (Figure 3). The difference may reflect the competing limitations of light and iron availability on the evolution of phytoplankton chl-a (e.g., Arteaga et al., 2020) and the associated variations in phytoplankton bloom phenology across a relatively large geographical area (Sallée et al., 2015; Turner et al., 2024) in the Southern Ocean. In the Northern Hemisphere, which includes the Arctic Ocean, phytoplankton growth is closely related to the retreat of sea ice and the subsequent availability of light. Macronutrients (nitrate, phosphate and silicate) are rapidly depleted by phytoplankton growth in the sunlit layer during the spring bloom and remain depleted until wintertime mixing replenishes these from deeper waters (e.g., Manizza et al., 2023), at which time phytoplankton growth becomes light limited (Ardyna and Arrigo, 2020). These limitations at the onset of winter may produce the steeper decline in chl-a concentrations in the Northern Hemisphere, from the spring- and autumntime chl-a. The relationships determined from the BGC-Argo profilers would therefore appear consistent with our understanding of the seasonal variability in phytoplankton.

Interannual variability in the response of the wintertime chl-a was apparent, particularly in the Southern Hemisphere (Figure 4d, e). Although the observation-based approach developed here does appear to capture interannual variability in the winter chl-a response, it is potentially underestimated. Reasons for this include the possibility that the BGC-Argo relationship may not fully capture the interannual variability between the autumn or springtime chl-a and the wintertime response. Alternatively, the mean interannual relationship will inherently be weighted towards the years (and their conditions) in which more BGC-Argo profiles were available during winter, i.e., the 2014, 2018 and 2024 periods (Figure 2). This uneven sampling of the BGC-Argo profilers could have a larger impact in regions that are experiencing rapid changes, for example the Arctic Ocean which has declining sea ice concentrations and increasing primary production (e.g., Lewis et al., 2020). We

do not see this as a limitation of the gap filling method as the differences are likely to be captured within the calculated uncertainties. As the BGC-Argo network reaches the intended ~1,000 profilers (Roemmich et al., 2019; although not all of these could have chl-a sensors), the interannual differences in the BGC-Argo chl-a wintertime relationships could be further investigated. However, at the time of writing there are ~700 BGC-Argo floats globally, of which ~400 floats have chl-a sensors.

The exploitation of satellite-based chl-a data records within, for example, ocean $CO_2$ sink assessments, is currently hampered by the missing data due to both cloud cover and in polar regions during wintertime. Within these assessments, the biological component has been shown to be an important predictor variable in approaches to estimate the in-water $CO_2$ concentrations (Ford et al., 2022). However, currently to exploit chl-a (or primary production) data within these assessments the missing

polar winter data are filled with fixed chl-a concentrations. For example, Gregor and Gruber (2021) set a fixed value of 0.3 mg m$^{-3}$ (blue dashed line in Figure 5). Here, the results show that the use of fixed values for wintertime chl-a concentrations overlooks the regional variability in wintertime chl-a and can in some cases lead to an elevated chl-a concentration above that of the spring bloom during wintertime (Figure 5e, f). These fixed values will also not capture the interannual variability in the wintertime data, which could lead to discontinuities in the chl-a data record and the predicted in-water $CO_2$

concentrations. The approach and resulting data presented here could therefore be used for these studies as it provides a gap-filled chl-a data record that is consistent with the underlying ocean colour satellite climate data record.

Although we advocate for dynamic values and using observation-based approaches, the under-ice regions were filled with a fixed value of 0.1 mg m$^{-3}$. The observational chl-a information under ice coupled with no availability of satellite ocean-

430 colour data inherently limits our ability to assign a dynamic value for these under-ice regions. Many studies have identified localised under ice phytoplankton blooms that can reach chl-a greater than 1 mg m$^{-3}$ (Arrigo et al., 2012, 2014; Boles et al., 2020), but their prevalence over the larger synoptic scales is unclear. Capturing these dynamics will rely on further advances in our understanding of these under-ice environments. BGC-Argo profilers have been deployed with ice avoidance systems (Randelhoff et al., 2020), which could be more widely deployed in sea ice regions to provide an in situ constraint to these

435 under ice environments. Within our approach, the choice of a fixed under-ice chl-a value will have different effects depending on the application. But it likely has limited effect on the ocean $CO_2$ sink approaches previously mentioned due to the current assumptions of no $CO_2$ exchange occurring in regions of high ice concentrations (see references within Watts et al., 2022).

The OC-CCI provides a climate-quality and consistently produced chl-a data record that performs well at the global scale (Sathyendranath et al., 2019). For our approach, the BGC-Argo chl-a were bias-corrected to the OC-CCI observations to maintain the consistency within the wintertime gap filled values, with respect to the observational record. Although the approach could be applied without first bias-correcting the BGC-Argo chl-a, this would introduce a bias 'step' that could

impact the retrieval of the seasonal cycle and the determination of trends within the record (Van Oostende et al., 2022). A

bias 'step' would reduce the consistency with the OC-CCI observations and the whole record which is a key component of a climate data record. The ability to produce consistent wintertime data with the underlying satellite data record (Figure 2 and 3) also allows the methods to be transferred to other data records (e.g., Hong et al., 2023).

Our results would indicate further *in situ* observations within the Southern Hemisphere and North Pacific Ocean would

improve the ability to incorporate and assess the accuracy of approaches for estimating wintertime chl-a concentrations. But equally, it reinforces the need for continued development of the OC-CCI record, and underlying chl-a retrieval algorithms (e.g., O'Reilly & Werdell, 2019).

## 5. Conclusion

In this study, we present an observation-based approach to fill gaps in the polar wintertime chl-a satellite data using BGC-Argo profiler observations. We apply the approach, alongside a cloud gap filling approach based on spatial kriging, to monthly 0.25 ° composites of chl-a from the Ocean Colour Climate Change Initiative (OC-CCI) climate data record to produce a gap-filled and spatially complete data record between 1997 and 2024 along with the propagated uncertainties. Data from BGC-Argo profilers during the polar winter were used to construct relationships between the wintertime chl-a and

the last available autumntime and first available springtime satellite observations. The BGC-Argo based gap filling approach retains the accuracy of the underlying dataset, as assessed with independent *in situ* observations, to produce a coherent timeseries. The resulting data identifies biogeochemical differences in the wintertime chl-a response between the Southern and Northern Hemispheres, whereby the Northern Hemisphere showed a faster and larger decline in chl-a than that in the Southern Hemisphere. These differences appear consistent with our understanding surrounding the seasonality of

phytoplankton in these biogeochemical different regions.

Applying the polar winter gap filling approach indicated that the gap-filled timeseries correctly captures the wintertime decline in chl-a and the interannual variability in the wintertime chl-a. The regional variability in the wintertime chl-a illustrated that the use of fixed values (as often used in the literature) to fill polar wintertime data is likely unsuitable and will result in misleading analyses and could even result in wintertime chl-a concentrations higher than those observed during the

spring when concentrations peak. The gap filling approach could be applied to any satellite based chl-a timeseries, and theoretically, for any biogeochemical variable that displays a similar wintertime response (e.g., particulate organic carbon or primary production). This study therefore provides a gap-filled coherent timeseries that can be exploited by communities that require spatially complete, gap-filled timeseries, for example as needed by machine learning approaches.

## Acknowledgements

This work was funded by the European Space Agency under the projects 'Satellite-based observations of Carbon in the Ocean: Pools, Fluxes and Exchanges' (SCOPE; 4000142532/23/I-DT) and 'Ocean Carbon for Climate' (OC4C; 3-18399/24/I-NB). This work was also partially funded by OceanICU which was funded by the European Union under grant agreement no. 101083922 and UK Research and Innovation (UKRI) under the UK government's Horizon Europe funding guarantee [grant number 10054454, 1006367, 10064020, 10059241, 10079684, 10059012, 10048179]. Views and opinions expressed are however those of the author(s) only and do not necessarily reflect those of the European Union or European Research Executive Agency. Neither the European Union nor the granting authority can be held responsible for them. This work is a contribution to the activities of the National Centre of Earth Observation (NCEO) of the United Kingdom. We also acknowledge additional support from the Simons Collaboration on Computational Biogeochemical Modelling of Marine Ecosystems/CBIOMES (Grant ID: 549947, SS). For the purpose of open access, the authors have applied a Creative Commons Attribution (CC BY) licence to any Author Accepted Manuscript version arising from this submission.

BGC-Argo data were collected and made freely available by the International Argo Program and the national programmes that contribute to it. (https://argo.ucsd.edu, https://www.ocean-ops.org). The Argo Program is part of the Global Ocean Observing System (GOOS). BGC-Argo data were collected and made freely available by the Southern Ocean Carbon and Climate Observations and Modeling (SOCCOM) project funded by the National Science Foundation, Division of Polar Programs (NSF PLR -1425989 and OPP-1936222 and 2332379), supplemented by NASA, and by the International Argo Program and the NOAA programs that contribute to it (http://www.argo.ucsd.edu, https://www.ocean-ops.org/board).

## Data and Code Availability

The monthly 0.25º gap-filled OC-CCI record are available from Zenodo (Ford et al., 2025b). The OC-CCI climate data record (v6) were retrieved from https://doi.org/10.5285/5011D22AAE5A4671B0CBC7D05C56C4F0 (Sathyendranath et al., 2023a), and the 2023 extension from https://www.oceancolour.org/. The OSI-SAF sea ice concentrations (v3) were retrieved from https://doi.org/10.15770/EUM_SAF_OSI_0013 (OSI SAF, 2022). The BGC-Argo data are available from the Argo GDAC, and the snapshot used within this work is available from https://doi.org/10.17882/42182#121877 (Argo, 2025). The code supporting this work is available at https://github.com/JamieLab/SCOPE-ArgoChla and archived at Ford et al. (2025a).

## Competing Interests

The authors declare no competing interests.

## Author Contributions

DJF and JDS conceived the study. DJF developed the methodology with inputs from JDS, GK and SS. DJF wrote the
505 original draft. All authors contributed to reviewing and editing the manuscript.

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
