# Peer review of "Decadal and spatially complete global surface chlorophyll-a data record from satellite and BGC-Argo observations"

_Earth System Science Data, 2025_

## Author Response (AR1)

Dear Editor and Reviewers,

We thank you for the reviews of our manuscript. We greatly appreciate receiving these detailed and constructive reviews and they have helped us to improve the paper considerably. Line numbers within this document refer to the tracked change version of the manuscript.

Within the review process we have taken the opportunity to update the dataset to include the year 2024, so the period 1997 to 2024 is now covered, and this led to updating our ingested BGC-Argo data to cover the longer period (last data ingestion was the 8th September 2025). The Zenodo repository has been updated with the latest version of our dataset (v0-3).

Yours sincerely,

Daniel J. Ford
* * *
Reviewer Comment 1 (RC1)

This manuscript aims to reconstruct global satellite Chlorophyll a (Chl) fields over the last 25 years, combining satellite observations with BGC-Argo float data and spatial kriging techniques. Gap-free Chl fields, including polar regions during wintertime, are vital for applications that require complete inputs, such as ocean CO2 sink assessment. This paper is well-motivated and enjoyable to read, especially the discussion section, which was strong and well-argued. The associated dataset on Zenodo is very well-referenced and is likely to be useful for the community.

That said, I have several concerns and suggestions, notably on some of the methods employed that I believe should be addressed before publication.

**Response:** *We thank the reviewer for their appraisal of our manuscript and have addressed all of their comments and suggestions below.*

General comments :

My main comment about this manuscript is that it foregoes an important limitation of satellite-based Chl reconstructions: photoacclimation under clouds. By computing monthly averages from only clear-sky days, one neglects the increased intracellular Chlorophyll content of phytoplankton under reduced light under clouds. This increase can achieve a tenfold factor. Consequently, reconstructions based solely on clear-sky spatial patterns (whether kriging, DINEOF, or others) underestimate Chl in regions with significant cloud coverage. While I recognize this limitation may be beyond the scope of the present study, I strongly suggest including a discussion of photoacclimation as a key caveat for Chl gap-filling.

**Response:** *We have now included a discussion on the effect of photoacclimation under clouds, and highlight the effect is beyond the scope of this study. As noted by the reviewer this limitation affects all products that use ocean colour records to reconstruct chl-a, and therefore is not unique to this study. The new text at Lines 389-396 reads as "It is important to note that spatial gap filling methods make assumptions about the missing data and use chl-a observations from clear sky conditions to fill these gaps. This will likely lead to an underestimation of chl-a concentrations due to photoacclimation by phytoplankton under reduced light from persistent cloud cover (i.e., increasing intracellular chl-a due to lower light conditions) (Begouen Demeaux et al., 2025). The construction of monthly composites of chl-a from observations in clear sky conditions could lead to a varying underestimation of chl-a based on regional cloudiness, for example subtropical gyres are likely less effected due to*

*persistent atmospheric high pressure. This limitation is not unique to this study as it will affect any ocean colour chl-a data product (e.g., Hong et al., 2023; Saulquin et al., 2018) and we therefore consider this outside the scope of the present study.".*

I would also moderate some of your conclusions about the Wintertime gap-filling. Although there is a relatively good fit between the Wintertime gap filled and the Valente HPLC data, all other comparisons, by any metrics employed, are not what I would qualify as "good". I believe the results against fluorometric data warrant a discussion on sources of disagreement and what further steps can be used to reduce this gap. Integrating over the whole penetration depth rather than 20meters and changing the technique used for the fluorometric correction would likely improve the performance of the reconstructions, see details below.

***Response:*** *We thank the reviewer for these suggestions. We have now addressed the methodological suggestions in their respective comments below.*

From personal preference, throughout this manuscript (and in the figures), I would consider using linear scale units for the reader to relate to the Chl values present. In the figures, I would suggest putting the Chl values in mg m-3 and using a logarithmic axis for increased readability.

***Response:*** *We have now modified the figures and text within the manuscript to have chl-a in units of mg m$^{-3}$. However when discussing the uncertainties, to remain consistent to the underlying OC-CCI dataset, we have kept the use of logarithmic units.*

Specific comments:

BGC-Argo usage: Although BGC-Argo floats are a formidable tool to validate and complement remote sensing data, some slight methodological changes could result in an improved product.

BGC-Argo data estimate Chl from fluorescence, which is not directly relatable to Chl from Satellite. In Roesler (2017), which you cite, they identify an average factor of 2 difference, which has a large variability across regions (up to a factor of 6 in the Southern Ocean !). In Section 2.4 you mention that you do account for this, but I don't believe that the method employed is accurate. The Roesler paper does not suggest using a single value bias but rather using a "Slope factor", which is much more accurate than a single bias value (that affects very differently small and large Chl concentrations). Accurately applying a Slope factor should significantly improve the relationship between BGC-Argo and OC-CCI (See Xing (2011) on a method to compute it from float radiometry). It is unclear to me if Figure S1 is prior or after the fluorescence-bias correction, but I would expect it to be much closer to a 1:1 line after a slope factor correction and reduce your "high intercept values" that you mention in paragraph 115.

***Response:*** *We thank the reviewer for their suggestions, which we have now implemented within the processing. The BGC-Argo delayed mode processing applies a slope correction* (Schmechtig et al., 2023) *as described in Roesler et al.* (2017)*, but using the average factor of 2 (as the reviewer highlights). This likely explains why our original fixed bias correction in the Arctic Ocean was small (0.02 log$_{10}$ (mgm$^{-3}$)), and we required a larger bias correction for the Southern Ocean (0.30 log$_{10}$ (mgm$^{-3}$)).*

*We have now implemented the slope correction (instead of the fixed bias correction) and identify a median slope correction of 0.916 for the Arctic Ocean and 1.967 for the Southern*

*Ocean. We highlight that these are applied after the BGC-Argo delayed mode processing has already applied a slope factor of 2. This is why the Arctic Ocean has a factor slightly less than 1, and the Southern Ocean had a factor of ~2. These corrections would be in the range identified in Roesler et al. (2017) and Long et al. (2024). This information is now found in the text at Lines 136-145 and reads "To minimise these differences, the BGC-Argo chl-a were corrected with respect to the OC-CCI data where coincident observations were available by using a slope factor correction as outlined in Roesler et al. (2017) (i.e., the median of all individual slope factors for each hemisphere). The Northern and Southern Hemisphere slope factor corrections were 0.916 and 1.967, respectively. These corrections appear initially smaller than those reported in Roesler et al. (2017) and Long et al. (2024), however the delayed mode processing of the BGC-Argo chl-a already includes a slope factor correction of 2. Therefore our slope factors are consistent to the previous work. The slope factor correction does not assume that the OC-CCI record is the 'truth', but our objective is to fill gaps in the OC-CCI data using relative changes to the BGC-Argo chl-a, which requires the two datasets to be consistent.".*

*Figure S1 has been updated to include the uncorrected and corrected BGC-Argo data, where the slopes look identical as the correction is applied to the untransformed chl-a data. The updated Figure S1 can be found below:*

[Figure]

***Figure S1:*** *(a) Comparison between daily matched Ocean Colour Climate Change Initiative (OC-CCI; 4km) and BGC-Argo chlorophyll-a (chl-a) within the Southern Hemisphere. Blue points indicate the uncorrected BGC-Argo data, and red points indicate the slope-corrected BGC-Argo data. Solid line is 1:1, and dashed blue line indicates a Type-II linear regression. The statistics are for the uncorrected BGC-Argo data, and acronyms are root mean square difference (RMSD) and number of samples (N). (b) same as (a), but for the Northern Hemisphere. (c) same as (a) but using monthly 0.25° composites of OC-CCI chl-a and BGC-Argo observations averaged to the same grid for the Southern Hemisphere. (d) same as (c), but for the Northern Hemisphere.*

You explain computing the mean Chl value from the first 20meters of each profile. This likely underestimates your Chl compared to what the Satellite measures, as increased chlorophyll concentration (such as in the DCM) can be found at depths deeper than 20meters, yet within the layer visible from the satellite. When comparing Satellite data with in-situ profiles, a commonly accepted technique is, for a given profile, to integrate/average over the penetration depth (Zpd) as this is considered a good approximation of what the satellite sees. It is computed as Zpd = 1/Kd(490). I would suggest retrieving Kd(490) either from the

Satellite pixel or, for more accuracy, to compute it from a BGC-Argo float Ed(490) profile, see Xing (2020).

*Response: We thank the reviewer for this great suggestion. The BGC-Argo floats could not all be matched to satellite $K_d(490)$, and not all of the floats used carried downwelling irradiance sensors. We therefore could not use either of these techniques to estimate the penetration or optical depth for each BGC-Argo profile. Instead we have estimated the $K_d(490)$ using the shallowest chl-a observation from each BGC-Argo profile (with a quality flag of 2), and the relationship between chl-a and Kd(490) described in Morel et al. (2007).*

*Although using the BGC-Argo chl-a may overestimate the optical depth, due to the underestimation of chl-a with respect to the satellite data, the approach does allow the optical depth to vary. Additionally, we have applied a lower limit to the BGC-Argo chl-a that we consider within the analysis as described in Long et al. (2024). This lower limit is 0.014 mg m$^{-3}$, which is twice the factory-specified sensitivity of the fluorescence sensors.*

*The modification to the methodology has now been included at Lines 88-94 which reads "For each BGC-Argo profile the quality flagging was applied to only retain the highest quality data (quality flag 2). The mean chl-a concentration was extracted from the first optical depth. The first optical depth was estimated from the diffuse attenuation coefficient at 490 nm ($K_d(490)$) which was determined using the shallowest chl-a observation (shallower than 10 m) and the relationship described in Morel et al. (2007). The mean was calculated in $log_{10}$ space due to the logarithmic distribution of chl-a (Campbell et al., 2002). Profiles with a mean chl-a less than 0.014 mg m$^{-3}$ were discarded as this value was twice the factory-specified sensitivity of the fluorescence sensors (Long et al., 2024).".*

Some of the information on the correction that is in 2.4 would probably be more appropriate in Section 2.2, so the reader knows at once how the BGC-Argo data were processed.

*Response: As suggested, we have moved the information on the slope factor correction to Section 2.2, at Lines 132-145.*

On the Spatial Kriging:

Please quantify the fraction of ocean pixels filled by kriging versus BGC-Argo. Supplementary Fig. S3 shows temporal coverage, but a spatial map distinguishing contributions of each method would be more informative. Additionally, it is important to mention that this Kriging method is effectively not filling specifically cloudy values but rather any pixels that have been permanently obscured for a given month. This should be emphasized, as persistent coccolithophore blooms have also resulted in pixels being flagged. I understand that most empty pixels are caused by clouds, but the technique here is not specific to clouds.

*Response: We have now added text that the data filled with the spatial kriging approach are not only cloud gaps, but could be due to other features inhibiting the satellite retrieval of chl-a. This text reads at Lines 162-164 as "But aggregating data over multiple days cannot help in regions where clouds can be more prevalent, such as the inter-tropical convergence zone, or in regions where other features, such as coccolithophore blooms, inhibit the satellite retrieval of chl-a.".*

There are also numerous BGC-Argo floats and in-situ datapoints from the Valente dataset in the area filled by the spatial Kriging. Although the paper's main point is not on this already

published method, it would be strengthened by the evaluation of the performance of the Spatial Kriging.

***Response:*** *We have now added a new supplementary figure (Figure S4) that replicates Figure 6 but for locations that have been filled by the spatial kriging approach. This figure shows relatively good performance of the cloud kriged chl-a against both the HPLC (Figure S4a) and fluorometric chl-a (Figure S4b). This figure is now referred to within the text at Lines 377-382 which reads as "In this study the use of monthly composites does reduce the number of observations that need to be gap-filled by the cloud kriging approach (Supplementary Figure S3), and therefore the computational cost of more complex methodologies likely outweighs any benefit to the retrieved chl-a. This situation is confirmed as the chl-a in regions where gaps were filled using the spatial kriging approach showed good performance with respect to the independent in situ observations (Supplementary Figure S4).".*

*Figure S4 is shown below.*

[Figure]

***Figure S4:*** *(a) Comparison between monthly 0.25°-gridded in situ HPLC chl-a concentrations and the gap-filled chl-a data using a spatial kriging approach. Solid line is 1:1 and dashed blue line indicates a Type-II linear regression. In text the abbreviations for the statistical measures are root mean square difference (RMSD) and number of samples (N). (b) Comparison between monthly 0.25°-gridded in situ fluorometric chl-a concentrations and the gap-filled chl-a.*

On the wintertime reconstruction:

In general, I thought the wintertime BGC-Argo reconstruction method could benefit from additional details, some reorganizing, and perhaps a schematic? I had to reread this section several times, and I am still convinced I have not understood this section correctly.

The first BGC-Argo floats used in this study were deployed in 2008, and yet Figure S2 and your text mention that the maximum time lag between OC-CCI and BGC-Argo used to fill a gap was 9 months. I am therefore unclear on how the Wintertime reconstruction was performed for those 9 years before the first Argo profile? From looking at Figure S2, I am hypothesizing that you used the BGC-Argo float to create some kind of monthly climatology in pixels, but I was unable to find explicit mention of this in Section 2.4. You mention "For each time lag, the median percentage difference was calculated between the OC-CCI and

bias-corrected BGC-Argo chl-a in mg m-3 […] on a pixel-by-pixel basis". I am again assuming you mean across all 25 years, within a given pixel, you find all BGC-Argo profiles that occurred within a month of the last OC-CCI measurement and compute the median, before repeating this for the 2-month lag and so on, but I believe the reader would really benefit from a rewriting of this section.

*Response: We thank the reviewer for the great suggestion and have now created a schematic of the whole methodology, which is now Figure 1. The new Figure 1 is displayed below this response. We have reorganised and restructured Section 2.4 to make the wintertime gap filling methodology clearer. The updated text can be found in Section 2.4 at Lines 184-224.*

[Figure]

*Figure 1: Schematic showing the methodology for producing the gap filled chlorophyll-a (chl-a) Ocean Colour Climate Change initiative (OC-CCI) record. In flowchart acronyms are Biogeochemical Argo (BGC-Argo), Argo Global Data Assembly Centers (GDAC), diffuse attenuation coefficient at 490 nm ($K_d(490)$) and Ocean and Sea Ice Satellite Application Facility (OSISAF).*

I would put emphasis in the discussion that the wintertime reconstruction is based on data from 2008 on and acknowledge the fact that this reconstruction is weighted around the time period in which there are more floats. This information is presented in Figure 1, but the limitations associated with this technique and uneven sampling frequency is only quickly mentioned in paragraph 365, and would benefit from a more thorough discussion, linking it to areas that have experienced rapid change in productivity and ice coverage over the last 10 years, notably in the Arctic.

*Response: We have now included further discussion highlighting that the effect of the uneven sampling likely impacts regions experience more rapid change such as the Arctic Ocean. This new text reads at Lines 432-438 as "Alternatively, the mean interannual relationship will inherently be weighted towards the years (and their conditions) in which more BGC-Argo profiles were available during winter, i.e., the 2014, 2018 and 2024 periods (Figure 2). This uneven sampling of the BGC-Argo profilers could have a larger impact in*

*regions that are experiencing rapid changes, for example the Arctic Ocean which has declining sea ice concentrations and increasing primary production (e.g., Lewis et al., 2020). We do not see this as a limitation of the gap filling method as the differences are likely to be captured within the calculated uncertainties.".*

*In the original manuscript, we discussed how the expanding availability of BGC-Argo profilers with chl-a sensors would allow the exploration of interannual variability in the wintertime relationships at Lines 438-441. Figure 2 would indicate in the more recent years there is an opportunity to start exploring these differences in future work.*

In Figure 2, the boxplots show a very large spread in the percent difference. It would be interesting to see if there are spatial patterns around this spread, notably if some areas of the Northern Hemisphere decrease in Chl more rapidly than others.

**Response:** *We have now plotted the data used to generate the percentage differences in Figure 3 for both the autumntime (backwards; Figure R1) and springtime (forwards; Figure R2) geographically for each month lag. Figures R1 and R2 are included below in this response document. Although we see some weak geographical differences visually, the large spread in percent differences appears to mainly stem from the comparison of BGC-Argo and OC-CCI at coincident locations (and this information is shown in Supplementary Figure S1). These plots did not highlight additional information and therefore we have not included them in the manuscript or made changes to the text. We hope the reviewer understands our reasoning.*

[Figure]

**Figure R1:** *(a) Percentage difference between BGC-Argo chl-a and OC-CCI for the autumntime (backwards) relationship at time lag 0. (b) to (j) same as (a) at the month lag highlighted above each subplot.*

**Figure R2:** *(a) Percentage difference between BGC-Argo chl-a and OC-CCI for the springtime (forwards) relationship at time lag 0. (b) to (j) same as (a) at the month lag highlighted above each subplot.*

For the in-situ Valente data of Chl fluorescence, has a conversion been applied similarly to the fluorescence by Argo (the Slope factor from Roesler, (2017)) ?  This might significantly help improve the comparison with Chl reconstructed from BGC-Argo.

*Response:* The in situ data from Valente et al. (2022) *did not have a slope factor applied as all of these in situ observations were made using in vitro (based on filtered water) fluorometric or spectrometric techniques. Valente et al. (2022) explicitly did not include observations made by in vivo fluorescence measurements from CTD sensors (similar to the BGC-Argo sensors) due to the potential problems with calibration. Figure 6 in Valente et al. (2022) shows good comparison between HPLC and fluorometric chl-a at stations with both observations, although a ~0.1 $\log_{10}(mg\ m^{-3})$ overestimation by the fluorometric chl-a was observed.*

*Within the revisions process, we identified an issue within our application of the quality flagging of the Valente et al. (2022) dataset, which has now been corrected. The updated Figure 6 in this study now shows a better performance (due to the very low chl-a values being removed within the flagging process), and the observed bias between OC-CCI and fluorometric observations in both hemispheres are relatively consistent with Valente et al. . (2022). The updated Figure 6 is shown below.*

[Figure]

**Figure 6:** *(a) Comparison between monthly 0.25°-gridded in situ HPLC chl-a concentrations and the wintertime gap-filled chl-a data in the Northern Hemisphere. Solid line is 1:1 and dashed blue line indicates a Type-II linear regression. In text the abbreviations for the*

*statistical measures are root mean square difference (RMSD) and number of samples (N). (b) Comparison between monthly 0.25°-gridded in situ fluorometric chl-a concentrations and the wintertime gap-filled chl-a in the Northern Hemisphere. (c) Same as (a), but for the Southern Hemisphere. (d) same as (b), but for the Southern Hemisphere.*

Figure 3(b)-(f) It is impossible to distinguish between the two timeseries on the plots due to space constrain (the x-axis is squished). I would suggest either removing 1or 2 timeseries graph or splitting them over two rows, as currently I cannot draw any conclusion from those.

***Response:*** *We have now modified Figure 4 to include only 4 timeseries (instead of 5) and split these across two rows. The y axes are now common across adjacent plots. The updated Figure 4 can be found below.*

[Figure]

***Figure 4:*** *(a) Chl-a timeseries extracted for the location marked by the arrow between 1997 and 2024 (plotted using consistent y axes). Blue line indicates the OC-CCI timeseries without gap filling and black lines indicate the gap filled data. Grey shaded region indicates the 1σ uncertainty in chlorophyll-a. (b), (d), (e) same as (a) but for their respective locations. (c) Ocean Colour Climate Change Initiative (OC-CCI) chlorophyll-a (chl-a) with gap filling*

*approach applied to the full timeseries climatology (1997 – 2024). We note that areas with a sea ice coverage greater than 90% are set to a fixed value of 0.1 mg m-3. Basemap in (c) from Natural Earth v4.0.0 (https://www.naturalearthdata.com/).*

Figure 3 and 4: I would put the time series values back in linear scale unit.

**Response:** *We have now modified the figures as suggested to have a linear scale unit, instead of the log transformed versions. The updated Figure 4 can be found above this response, and the updated Figure 5 is below this response.*

[Figure]

**Figure 5:** *(a) Ocean Colour Climate Change Initiative (OC-CCI) chlorophyll-a (chl-a) with gap filling approach applied to the full timeseries climatology (1997 – 2024). We note that areas with a sea ice coverage greater than 90% are set to a fixed value of 0.1 mg m-3. (b) Monthly climatology calculated at the location marked by the arrow. Blue line indicates the monthly climatology for the OC-CCI timeseries. Black line indicates the monthly climatology for the gap-filled OC-CCI, where the grey shading indicates one standard deviation of the gap-filled climatology. Dashed blue line indicates a chl-a value of 0.3 mg m-3 and is referred to in the text. (c), (d), (e) and (f) same as (b), but for their respective locations. Basemap in (a) from Natural Earth v4.0.0 (https://www.naturalearthdata.com/).*

Roesler, C., Uitz, J., Claustre, H., Boss, E., Xing, X., Organelli, E., Briggs, N., Bricaud, A., Schmechtig, C., Poteau, A., D'Ortenzio, F., Ras, J., Drapeau, S., Haëntjens, N., & Barbieux, M. (2017). Recommendations for obtaining unbiased chlorophyll estimates from in situ chlorophyll fluorometers: A global analysis of WET Labs ECO sensors. Limnology and Oceanography: Methods, 15(6), 572–585. https://doi.org/10.1002/lom3.10185

Xing, X., Morel, A., Claustre, H., Antoine, D., D'Ortenzio, F., Poteau, A., & Mignot, A. (2011). Combined processing and mutual interpretation of radiometry and fluorimetry from autonomous profiling Bio-Argo floats: Chlorophyll a retrieval. Journal of Geophysical Research, 116(C6), C06020. https://doi.org/10.1029/2010JC006899

Xing, X., Boss, E., Zhang, J., & Chai, F. (2020). Evaluation of Ocean Color Remote Sensing Algorithms for Diffuse Attenuation Coefficients and Optical Depths with Data Collected on BGC-Argo Floats. Remote Sensing, 12(15), 2367. https://doi.org/10.3390/rs12152367
* * *
Reviewer Comment 2 (RC2)

This paper proposes a procedure to generate a gap-free time series of global monthly chlorophyll maps based on satellite data. The novelty is that this product covers also the polar oceans thanks to the availability of bio-argo profiles during the polar night. This effort is motivated by the common need in climate studies for spatially complete datasets spanning the longest possible time interval. The time series begins in 1998 with the onset of ocean colour missions and is openly available on Zenodo, representing a valuable resource for numerous climate research applications.

Considering the usefulness and the importance of the proposed data set I am convinced that the paper merits to be published but, at the same time, I also have some major concern about the reconstruction methods applied, that need to be better explained and qualified. In addition, I am not convicted on the claimed better performance of the Kriging method respect other methods commonly used in literature or routinely applied by operational centres such as the CMEMS Ocean Colour Thematic Assembly Center (surprisingly is never cited in the manuscript).

The method used to reconstruct the chlorophyll field during the polar night, while rather convoluted and extremely crude, it is better or, at least, less wrong than use a constant value representing a small step forward in the production of global gap-free satellite images.

***Response:*** *We thank the reviewer for their positive appraisal of our manuscript, and we have now addressed all of your comments below.*

Specific Comments:

2.2 Satellite observational data:

1 - Daily maps used to produce monthly means are the result of a composition of several passage acquired by several satellite missions. Can you report here how the daily aggregated maps are produced or, at least include a citation where how the daily composite maps are produced is described?

***Response:*** *We have now included a sentence that highlights the reference for the OC-CCI processing chain which cross-calibrates and then merges the data from multiple satellite mission into daily composites, and then monthly composites. This text reads at Lines 105-108 as "The OC-CCI (v6) chl-a data were retrieved at daily and monthly temporal resolution at their native spatial resolution (4 km) between October 1997 and December 2023 (Sathyendranath et al., 2019, 2023a). The process for cross-calibrating and merging the data from multiple satellite ocean colour sensors used within the OC-CCI are described in Sathyendranath et al. (2019).".*

2 – line 100: "(1 sigma; given as the root mean square difference; RMSD)". RMSD of what?

***Response:*** *We have now clarified that the root mean square difference (RMSD) uncertainty estimate provided with the OC-CCI is calculated with respect to in situ observations within the optical water classes used in generating the OC-CCI record. This text reads at Lines*

*109-111 as "The uncertainties (1 sigma; given as the root mean square difference; RMSD) provided with the OC-CCI product are calculated with respect to in situ observations within each optical water class (Jackson et al., 2017).".*

3 – line 100-102. Also assuming that "spatial uncertainties within adjacent cells are dependent and spatially correlated" I am not totally sure that the mean of the 4 km uncertainties is the mean of the single standard deviation. Can you please indicate where in Taylor 1997 this specific point is discussed and proved?

*Response: These OC-CCI uncertainties were propagated through the calculation of the mean (i.e the mean of all 4 km observations within a 0.25 degree region), which involves the 4 km chl-a observations being summed and then divided by the number of observations. The uncertainties, as they are assumed dependent, will therefore also be summed (as discussed in Section 3.3 of Taylor 1997). They would then be divided by the number of observations, which is a constant with no uncertainty (as discussed in Section 3.4 of Taylor 1997).*

4 – line 113-115: The authors use a type II regression: are errors on BGC-Argo and OC-CCI data that enter in the regression comparable?

*Response: The use of a Type II regression acknowledges that both sources of data have an uncertainty. The OC-CCI data has a formal uncertainty (precision) per pixel that on a global scale equates to ~0.3 $\log_{10}$(mg m$^{-3}$). The BGC-Argo data does not have a formal uncertainty, and is still underdevelopment (see Section 4.4 of Schmechtig et al., 2023). As we highlight in the manuscript at Lines 69-71, the BGC-Argo data are generally lower accuracy then the OC-CCI (and we provide a correction for the accuracy difference), but currently it is not possible to identify the precision. We therefore have assumed that the precision of the two approaches is comparable in these analyses (whereas the alternative of using a Type I regression would assume that one dataset is truth, which would seem an impossible scenario within in situ or space observations).*

Section 2.3: Spatial Kriging for cloud gap filling

1 – line 130: ".... used to estimate the chl-a concentration",…  add citation.

*Response: We have now added references to this statement, which reads at Lines 157-158 as "Ocean-colour sensors on board polar orbiting satellites collect data at multiple wavebands in the visible domain, which is used to estimate the chl-a concentration (e.g. Gohin et al., 2002; O'Reilly and Werdell, 2019).".*

2 - The choice to reconstruct the field over data voids using Ordinary Kriging is primarily based on the work of Stock et al. (2020) which compares Ordinary Kriging, DINEOF, and several widely used AI methods. Optimal interpolation is not considered and other advanced methods based on Singular Spectra Analysis (Kondrashov, D., & Ghil, M., 2006 Spatio-temporal filling of missing points in geophysical data sets. Nonlinear Processes in Geophysics, 13(2), 151-159) are not even mentioned.  I understand that the Ordinary Kriging method is significantly less computationally demanding compared to some of more sophisticated methods (e.g. SSA or DINEOF or Optimal Interpolation); however, if this is the case, it should be clearly stated in the text, rather than simply claiming that Kriging and DINEOF perform better. it should be discussed and what is the advantage of using ordinary kriging respect to other kriging methods such as the method adopted by CMEMS L4 Global chlorophyll product (Saulquin et al, 2018).

In addition, it will be important to compare the proposed product with other monthly L4 global chlorophyll products such as those distributed by CMEMS (OCEANCOLOUR_GLO_BGC_L4_NRT_009_102, OCEANCOLOUR_GLO_BGC_L4_MY_009_104) and discuss the results.

*Response: We thank the reviewer for highlighting these points. We have now updated the text within the manuscript to address these more complex methods for filling the cloud cover gaps, but these will come with an increased computational cost. We now indicate that the spatial kriging was used because the use of monthly composites reduces the number of pixels that need to be filled by this method (as shown in Supplementary Figure S2), and therefore the additional computational cost in this setup is unlikely to improve the estimates further. This text reads at Lines 168-170 as "Here we implement a spatial ordinary kriging approach to fill cloud cover (or other) gaps (Figure 1) as the use of monthly composites in this study reduces the number of data gaps, and therefore the computational requirements of more complex approaches are unlikely to improve the estimates further.".*

*Within the discussion we have now added further text to highlight that these more complex methodologies, with their higher computational cost would likely have an impact on the retrieved chl-a when applying to higher temporal and spatial datasets. This text now reads at Lines 383-387 as "This may suggest that more complex methodologies, such as those proposed by Hong et al. (2023), using a convolution neural network (that considers the physical and biological conditions), could be more applicable to filling larger cloud cover gaps. With these larger gaps, the computational cost of these more complex methodologies could be beneficial in improving the retrieved chl-a.".*

*We thank the reviewer for the suggestion to compare our product to other L4 gap filled products such as the CMEMS Globcolour products. However, the underlying product generation between the CMEMS Globcolour and OC-CCI is inherently different. OC-CCI band shifts the reflectance at each wavelength from all input satellite sensors, merges the reflectance and then applies a chl-a algorithm to each optical water type (as described in Sathyendranath et al., 2019). The CMEMS Globcolour applies chl-a algorithms to the individual sensors and then merges these (Garnesson et al., 2019). We therefore would expect a difference in the retrieved chl-a for pixels where observations are available (e.g. Garnesson et al., 2019). These differences would propagate to any pixels or regions that are filled with the spatial kriging (in this study) or the optimum interpolation (from CMEMS) method and therefore would provide only limited information on the differences due to interpolation technique). Instead we have used the independent Valente et al. (2022) in situ chl-a to assess the performance at the cloud kriged locations, and this information is shown in Figure S4. Figure S4 is shown on Page 5 of this document. Figure S4 shows good performance for the cloud kriged locations against both the HPLC and fluorometric chl-a. This figure is referred to within the text at Lines 380-382 which reads "This situation is confirmed as the chl-a in regions where gaps were filled using the spatial kriging approach showed good performance with respect to the independent in situ observations (Supplementary Figure S4).".*

3 - How is polar night) anded? Below what solar elevation value is it defined as "polar night"?

*Response: The polar night in this study is defined using the underlying availability of OC-CCI observations. The OC-CCI processing means that ocean colour data where the solar zenith angle is greater than 70 ° are not processed further leading to the missing wintertime data. We apply the kriging approach within a band between the two latitudes where at least 20% of the OC-CCI observations are available, calculated on a monthly basis from the data.*

*Examples of these bands are presented in the new Figure 1, which is shown on Page 6 of this document, and the information was within the original manuscript and appears on Lines 173-175. The remaining locations with no data (and sea ice coverage less than 10%) are processed with the BGC-Argo wintertime relationships.*

4 – "…. The semi-variogram was fit to a ~5% subset of the OC-CCI observations that were equally distanced in space, for a monthly varying latitude band where at least 20% of the OC-CCI observations are available…." Does this mean that the parameters of the exponential function are calculated for each latitude band and, consequently, the fitted function depends on latitude? Please clarify.

**Response:** *The semi-variogram was fit to a ~5% subset of the OC-CCI observations in a latitude band where at least 20% of the OC-CCI observations were available, for example a 45 °N to 80 °S. This band was recalculated for each month within the timeseries. Within the new Figure 1 we have shown an example of this latitude band in the spatial kriging component of the flowchart, and how it changes between each month. The 5% observations used as 'tie points' are shown in the proceeding step, which then inform the spatial kriging to fill these gaps. Figure 1 is shown on Page 6 of this document. We have updated the text to clarify the information above, which reads at Lines 173-175 as "The semi-variogram was fit to a ~5% subset of the OC-CCI observations that were equally distanced in space, for a monthly varying latitude band (e.g. 50 °N to 80 °S; Figure 1) where at least 20% of the OC-CCI observations are available.".*

5 – Finally, considering that the cloud gap filling kriging approach uses observations in the vicinity of the empty pixel (see first line of the Uncertainty propagation section) is your kriging including a definition of an influential distance that limits the search radius?

**Response:** *The kriging approach does not have a defined influential distance, but it is limited to using the nearest six observations thereby limiting the influential distance. We have now included this information within the methods at Lines 179-181 which reads "The ordinary kriging was applied only to the missing data locations (i.e., the original OC-CCI observations are left unchanged) and was set to use the nearest six observations to fill a missing data location which limits the influential distance of each observation.".*

Section 2.4: BGC-Argo Wintertime filling

1 – In section 2.2, the authors correctly note that substantial differences can occur between OC-CCI (based on empirical algorithms that use HPLC data) and BGC-Argo chlorophyll measurements. However, in section 2.4, they address this issue by simply applying two constant bias corrections to the BGC-Argo data, one for each hemisphere, justifying this approach using the results shown in Figure S1.

Since in Figure S1 it is evident that the difference between the two dataset is not limited to bias, the question that arises here is: why not also account for the slope of the relationship, which would likely allow for a more accurate correction?

**Response:** *In response to both reviewers comments on the bias correction we have now performed a slope correction approach as described in Roesler et al. (2017). Using the slope factor we identify a median slope correction of 0.916 for the Arctic Ocean and 1.967 for the Southern Ocean. We highlight that these are applied after the BGC-Argo delayed mode processing has already applied a slope factor of 2. This is why the Arctic Ocean has a factor slightly less than 1, and the Southern Ocean had a factor of ~2. These corrections would be in the range identified in Roesler et al. (2017) and Long et al. (2024). This information is now*

*found in the text at Lines 136-145 and reads "To minimise these differences, the BGC-Argo chl-a were corrected with respect to the OC-CCI data where coincident observations were available by using a slope factor correction as outlined in Roesler et al. (2017) (i.e., the median of all individual slope factors for each hemisphere). The Northern and Southern Hemisphere slope factor corrections were 0.916 and 1.967, respectively. These corrections appear initially smaller than those reported in Roesler et al. (2017) and Long et al. (2024), however the delayed mode processing of the BGC-Argo chl-a already includes a slope factor correction of 2. Therefore our slope factors are consistent to the previous work. The slope factor correction does not assume that the OC-CCI record is the 'truth', but our objective is to fill gaps in the OC-CCI data using relative changes to the BGC-Argo chl-a, which requires the two datasets to be consistent.".*

2 – Figure 1b shows that before 2010 BGC-Argo profiles are not available. If I have correctly understood, the filling procedure adopted by authors requires to have satellite data in the next spring and/or in the previous autumn and BGC-Argo profiles in between. In the absence of BGC-Argo profiles, it is unclear how the filling procedure can be applied and how the data gaps are filled. Please clarify the methodology adopted.

*Response: We have now added a new schematic (the new Figure 1) that shows the full methodology used to construct the gap filled OC-CCI record and includes a section on the wintertime chl-a approach. The new Figure 1 is shown on Page 6 of this document. Section 2.4 has been restructured based on comments from both reviewers, showing that we construct the median percentage difference relationship between BGC-Argo and OC-CCI taking into consideration all of the available BGC-Argo profiles (i.e., a median of 2010 to 2024 percentage differences). The updated text appears at Lines 206-211.*

3 – line 179-181: "…..For each wintertime pixel, the time lag between the autumntime and springtime OC-CCI observations was calculated, and the relationship with the lowest time lag was used".

Since both values have been calculated, what prevents the use of a weighted average of the two chlorophyll values, in analogy with the approach used in the case of Kriging?

*Response: Although this could be implemented, this would add a further layer of complexity to the methodology that we feel would add little gain to the retrieved chl-a at this stage. The current implementation is easily traceable to which relationship has been used (and the time lag used), as this information is provided in the dataset netCDF files, and can be used to extract the OC-CCI observation used to estimate the wintertime observation. As we discuss for the cloud kriging approach (at Lines 166-167) the added complexity might provide a larger improvements when going to higher spatial and temporal resolutions. We hope the reviewer understands our reasoning for not implementing this within the analysis in the current study.*

4 – The authors wrote: "…..Any remaining pixels that were not gap filled by any of the previous procedures are filled with a final kriging pass……". In this regard, it would be useful to quantify the percentage of sea pixels that remain unfilled after applying the wintertime filling procedure.

*Response: The remaining unfilled pixels after applying the spatial kriging for filling the cloud gaps, and the wintertime filling procedure are restricted to the locations of partial sea ice coverage (i.e., those with 10% to 90% ice coverage). Supplementary Figure S3 has now been updated to show the percentage areal cover for each gap filling technique and shows*

*that ~3% of the data are filled with this final kriging pass, which has been added to the text at Lines 232-233. Supplementary Figure S3 is shown below.*

[Figure]

**Figure S3:** *The monthly percentage area contribution of pixels flagged by each gap filling approach between 1997 and 2024.*

Section 2.5: Uncertainty propagation

1 – lines 215-219: The description of the method used to estimate uncertainty in percentage differences is somewhat convoluted and difficult to follow. How is absolute deviation converted into an equivalent standard deviation? How are the two sources of uncertainty propagated through the analysis? And how is the Monte Carlo approach applied?

***Response:*** *We have now modified Section 2.5 to make the description clearer. To answer the reviewer's specific points. The median absolute deviation (MAD) was converted into an equivalent standard deviation (or a robust standard deviation) using the scaling factor of 1.4826 (Rousseeuw and Croux, 1993). This information is now on Lines 255-257 and reads "We estimate the uncertainty in the percentage difference by calculating the median absolute deviation (MAD) and convert this to a standard deviation equivalent with the scaling factor of 1.4826 (Rousseeuw and Croux, 1993).".*

*The two sources of uncertainty were propagated by randomly perturbing the input value (i.e., the percentage difference and the OC-CCI chl-a observation) within their uncertainties and recalculating the resulting wintertime chl-a 1000 times within the Monte Carlo. The standard deviation of the resulting 1000 ensemble was taken as the uncertainty on the wintertime chl-a and provided in $\log_{10}(mgm^{-3})$ to be consistent to the underlying OC-CCI record. This information is included at Lines 259-264 which reads "Each source of uncertainty was propagated by randomly perturbing the input value (i.e., the percentage difference and OC-CCI chl-a observation) using a random number generator that produces a normal distribution with a standard deviation defined by the uncertainty. The wintertime chl-a was then recalculated for each perturbed input in the ensemble. The standard deviation of the 1,000 ensembles was taken as the uncertainty and the resulting spatially varying uncertainty were provided in $\log_{10}(mg\ m^{-3})$ units for each of the polar wintertime filled pixels (to be consistent to the underlying OC-CCI record).".*

Section 3. Results

1 – Figure 3 shows a gap-free global map in which chlorophyll data are available at all latitude, Figure 1b shows that BGC-Argo profiles are available only below 75° N. It will be important to highlight in the figure caption that area covered by 90% of ice where set to 0.1 mg/m3.

***Response:*** *We have now added to the figure's captions that the high latitude ice covered areas have been set to 0.1 mg m$^{-3}$. This text reads at Line 347 and 354-355 as "We note that areas with a sea ice coverage greater than 90% are set to a fixed value of 0.1 mg m$^{-3}$.".*

Section 4. Discussion

1 - In the discussion line 317-319 it is recognized that produce higher temporal resolution datasets is possible but the results of the reconstruction over areas of large and persistent cloud cover could be questionable. While this is certainly correct, it raises the question of how much of the reconstruction difficulty is due to the use of purely spatial interpolation, as opposed to spatio-temporal interpolation employed in other approaches, such as Optimal Interpolation.

In addition, in this section will be important to compare the proposed product with other L4 available chlorophyll products used by the user community and discuss the difference and similarity.

***Response:*** *In response to a previous comment by the reviewer we have added further information on the choice for selecting the spatial kriging, which can be found at Lines 166-170 in the methods. In the discussion we have now added further information that the spatial kriging will likely struggle in periods of persistent cloud cover, where the added computational requirements of more complex interpolation methodologies will likely benefit the reconstructions when going to higher resolutions. This added text at Lines 377-387 reads "In this study the use of monthly composites does reduce the number of observations that need to be gap-filled by the cloud kriging approach (Supplementary Figure S3), and therefore the computational cost of more complex methodologies likely outweighs any benefit to the retrieved chl-a. This situation is confirmed as the chl-a in regions where gaps were filled using the spatial kriging approach showed good performance with respect to the independent in situ observations (Supplementary Figure S4). Applying the full methodology to generate a higher-temporal resolution dataset (e.g., by using 8-day composites instead of monthly composite) is possible but could present challenges when larger regions are covered by clouds. This may suggest that more complex methodologies, such as those proposed by Hong et al. (2023), using a convolution neural network (that considers the physical and biological conditions), could be more applicable to filling larger cloud cover gaps. With these larger gaps, the computational cost of these more complex methodologies could be beneficial in improving the retrieved chl-a.".*

*In a previous comment to the reviewer we highlight why the comparison of the data in this study to the CMEMS L4 product that uses optimum interpolation would highlight known differences in the underlying chl-a generation where ocean colour observations are available, and it would provide little information on the underlying gap filling technique (comment on Page 15 of this document). So instead, to provide an independent assessment of our work, we have used the independent Valente et al. (2022) in situ chl-a observations to assess the performance at the cloud kriged locations, and this information is shown in Figure S4. Figure S4 shows good performance for the cloud kriged locations against both the HPLC and fluorometric chl-a. Figure S4 is shown on Page 5 of the document. This figure is referred to within the text at Lines 380-382 which reads "This situation is confirmed as the chl-a in regions where gaps were filled using the spatial kriging approach showed good performance with respect to the independent in situ observations (Supplementary Figure S4).".*

References:

Saulquin, B., Gohin, F., and Fanton d'Andon, O.: Interpolated fields of satellite-derived multialgorithm chlorophyll-a estimates at global and European scales in the framework of the European Copernicus-Marine Environment Monitoring Service, J. Oper. Oceanogr., 12, 47–57, 2018, www.tandfonline.com/doi/full/10.1080/1755876X.2018.1552358.
* * *
Community Comment 1 (CC1)

I appreciate very much the ability to comment prior to publication. At the core of this work are the satellite data sets and the BGC Argo chlorophyll dataset. Likely all of the BGC Argo data used in this excellent paper come from the international OneArgo program. A large percentage of the BGC-Argo floats in OneArgo, larger than in core Argo, are funded by the US and specifically the US National Science Foundation. Most of the BGC Argo in the Southern Ocean are funded by NSF as part of the SOCCOM program. The Data and Acknowledments sections carry no references to Argo. This is a very simple and extremely vital correction, especially given the current grave threats to US funding for BGC Argo. Also, as would be understood by editors of a specifically data-oriented journal, detailed citations of data sources, not just an international compilation that eliminates attribution of the major funding and programmatic lift required to collect the data, only benefit all of us. I request that the Data statement include at least 2 acknowledgments, the first being the international Argo program and the second being the US NSF funded SOCCOM program. Both programs carry 'How to Cite' data statements on their websites.

The SOCCOM program has deployed 314 floats since 2014, all south of 30S and many in the sea ice zone. 144 are currently operational. (Lifetime is 4 to 5 years.)

The GO-BGC program has deployed 296 floats since 2021, with 51 currently south of 30S, enhancing the SOCCOM array and international BGC Argo array.

The attached screenshot from one of our recent presentations (at UNOC) shows the current contribution of SOCCOM and GO-BGC to the global BGC Argo array, and the graph shows the expected number of floats when the NSF ceases to fund acquisition of floats in November 2025 (this year). The US should be contributing about 500 total floats to the global BGC Argo array and will reach that contribution at the end of GO-BGC deployments (in the US NOAA is a very minor funder of BGC Argo). However, other nations are not yet contributing close to the total of 500 required for the complete global array.

OneArgo: https://argo.ucsd.edu/data/acknowledging-argo/ These data were collected and made freely available by the International Argo Program and the national programs that contribute to it. (https://argo.ucsd.edu, https://www.ocean-ops.org). The Argo Program is part of the Global Ocean

SOCCOM: https://soccom.org/about-us/acknowledgment-text/

"Data were collected and made freely available by the Southern Ocean Carbon and Climate Observations and Modeling (SOCCOM) Project funded by the National Science Foundation, Division of Polar Programs (NSF PLR -1425989 and OPP-1936222 and 2332379), supplemented by NASA, and by the International Argo Program and the NOAA programs that contribute to it. (http://www.argo.ucsd.edu, https://www.ocean-ops.org/board). The Argo Program is part of the Global Ocean Observing System."

*Response: We thank Lynne Talley for highlighting the omission of acknowledgements for the BGC-Argo data. We have updated the acknowledgements with the two acknowledgement statements as requested at Lines 516-521. We have now referenced the Argo database snapshot used within this study at Line 84-87, which reads "Delayed mode BGC-Argo profile data (2008 to 2024, last ingestion: 8th September 2025) (Argo, 2025) were retrieved from the Argo Global Data Assembly Centers (GDAC) for the Southern Hemisphere (south of 40°S) and the Northern Hemisphere (north of 40°N; Figure 1).". The Argo database is now also referenced within the data and code availability section at Lines 526-527.*
* * *
**References**

[revised manuscript text omitted]

---

## Referee Report (RR1)

**Review : Decadal and spatially complete global surface chlorophyll-a data record from satellite and BGC-Argo observations.**

This is my second round of review of this manuscript. I would like to commend the authors for their very thorough responses and for the care taken in addressing the reviewers' comments.

Significant (and I am sure time-consuming) changes were made in this version, including applying a slope factor to BGC measurements of Chlorophyll from Fluorescence. I would suggest showcasing the OC-CCI–Argo matchup statistics using the corrected Argo data in Figure S1.

The integration of chlorophyll to the first optical depth is also appreciated and strengthens the analysis.

The new Figure 1 is excellent and is very helpful to the reader.

Minor comments( at the author's discretion):

Figure 4: It is still quite hard to visualize the time series. I would suggest using a lighter shade of gray for the uncertainties.

Figure 5: Similar comment, the time series are hard to see.

Line 306 "Although, the precision is larger as highlighted by the RMSD values ~0.6 log10(mg m-3 ) compared to ~0.4 log10(mg m-3 ) for the comparison to the BGC-Argo chl-a (Supplementary Figure S1 d)." I am not sure I understand this sentence. Did you mean the error is larger?

Figure 5 and  Line 420 "For example, Gregor and Gruber (2021) set a fixed value of 0.3 mg m-3 (blue dashed line in Figure 5). Here, the results show that the use of fixed values for wintertime chl-a concentrations overlooks the regional variability in wintertime chl-a and can in some cases lead to an elevated chl-a concentration above that of the spring bloom during wintertime (Figure 5e, f)" . I do not see it in Figure 5.f., but in 5.b.

Could you expand on whether this difference is due to your under-ice fixed value of 0.1 mg m-3, or to the kriging?